# Trends and seasonal variability of ammonia across major biomes in Western and Central Africa inferred from long-term series of ground-based and satellite measurements

Money Ossohou[1,2]*, Jonathan Edward Hickman[3], Lieven Clarisse[4], Pierre-François Coheur[4], Martin Van Damme[4,5], Marcellin Adon[2,6], Véronique Yoboué[1], Eric Gardrat[7], Maria Dias Alvès[7] and Corinne Galy-Lacaux[7]

[1]Department of Physics, University of Man, Man, Côte d'Ivoire
[2]Laboratoire des Sciences de la Matière, de l'Environnement et de l'Energie Solaire, Université Félix Houphouët-Boigny, Abidjan, Côte d'Ivoire
[3]NASA Goddard Institute for Space Studies, USA
[4]Université Libre de Bruxelles (ULB), Spectroscopy, Quantum Chemistry and Atmospheric Remote Sensing (SQUARES), Brussels, Belgium
[5]Royal Belgian Institute for Space Aeronomy, Brussels, Belgium
[6]Laboratoire des Sciences et Techniques de l'Environnement, Université Jean Lorougnon Guédé, Daloa, Côte d'Ivoire
[7]Laboratoire d'Aérologie, Université Toulouse III Paul Sabatier, CNRS, France

*Correspondence to: Money Ossohou (ossohoumoney@gmail.com)

**Abstract.** Ammonia ($NH_3$) is the most abundant alkaline component in the atmosphere. Changes in $NH_3$ concentrations have important implications for atmospheric chemistry, air quality, and ecosystem integrity. We present a long-term ammonia ($NH_3$) assessment in the Western and Central Africa region within the framework of the International Network to study Deposition and Atmospheric chemistry in Africa (INDAAF) program. We analyze seasonal variations and trends of $NH_3$ concentrations and total columns densities along an African ecosystem transect spanning dry savannas in Banizoumbou, Niger and Katibougou, Mali, wet savannas in Djougou, Benin and Lamto, Côte d'Ivoire, and forests in Bomassa, Republic of Congo and Zoétélé, Cameroon. We use a 21-year record of observations (1998-2018) from INDAAF passive samplers and 11-year record of observations (2008-2018) of atmospheric vertical column densities from the Infrared Atmospheric Sounding Interferometer (IASI) to evaluate $NH_3$ ground-based concentrations and total column densities, respectively. Climatic data (air temperature, rainfall amount and leaf area index), as well as ammonia emission data of biomass combustion from the fourth version of the Global Fire Emissions Database (GFED4) and anthropogenic sources from the Community Emissions Data System (CEDS), were compared with total $NH_3$ concentrations and total columns over the same periods. Annual mean ground-based $NH_3$ concentrations are around 5.7-5.8 ppb in dry savannas, 3.5-4.7 ppb in wet savannas and 3.4-5.6 ppb in forests. Annual IASI $NH_3$ total column densities are 10.0-10.7 x $10^{15}$ molec $cm^{-2}$ in dry savanna, 16.0-20.9 x $10^{15}$ molec $cm^{-2}$ in wet savanna and 12.4-13.8 x $10^{15}$ molec $cm^{-2}$ in forest stations. Non-parametric statistical Mann-Kendall trend tests applied to annual data show that ground-based $NH_3$ concentrations increase at Bomassa (+2.56% $yr^{-1}$), but decrease at Zoétélé (-2.95% $yr^{-1}$) over the 21-year period. The 11-year period of IASI $NH_3$ total column density measurements show yearly increasing trends at Katibougou

(+3.46% yr$^{-1}$), Djougou (+2.24% yr$^{-1}$) and Zoétélé (+3.42% yr$^{-1}$). From the outcome of our investigation, we conclude that air temperature, leaf area index and rainfall combined with biomass burning, agricultural and residential activities are the key drivers of atmospheric NH$_3$ in the INDAAF stations. The results also show that the drivers of trends in the (1) dry savanna of Katibougou is agriculture, (2) wet savanna of Djougou and Lamto are air temperature and agriculture, and (3) forest of Bomassa are leaf area index, air temperature, residential and agriculture.

## 1 Introduction

Atmospheric nitrogen (N) compounds play an important role in all compartments of the critical zone (biosphere-atmosphere-hydrosphere) at the global scale. Since 2002, Bouwman et al. (2002a) had claimed that in the new future, both acidification and eutrophication risks due to excess of N could significantly increase in Asia, Africa and South America, but decrease in

North America and Western Europe. Reactive nitrogen (Nr) in the atmosphere, either reduced (NH$_x$ = NH$_3$ and NH$_4^+$) or oxidized (NO$_x$) forms, has a very different role. Ammonia (NH$_3$), the inorganic form of Nr typically produced through the deprotonation of NH$_4^+$, is the most abundant alkaline component in the atmosphere (Behera et al., 2013). In the atmosphere, NH$_3$ influences the abundance and chemical composition of sulfate particles, primarily from dimethyl sulfide (DMS) emissions arising from planktonic algae (Bouwman and Van Der Hoek, 1997). In the lower troposphere, NH$_3$ neutralizes a great portion

of the acids produced by oxides of sulfur and nitrogen (Adon et al., 2010) and forms fine particulate matter (PM$_{2.5}$) (Malm et al., 2004). Through wet or dry deposition to the surface, NH$_3$ can be detrimental over time due to an increased toxicity toward sensitive species of plants (Behera et al., 2013; Galloway et al., 2004), ecosystems (Erisman et al., 2013) and soils (Stevens et al., 2018). Different sources contribute to NH$_3$ emissions on the African continent, which in turn influence the seasonality of atmospheric concentrations and deposition of NH$_3$. Due to its high reactivity, a significant fraction of the NH$_3$ emitted is rapidly

deposited within a 1 km radius of the source (Fowler et al., 1998). It is clear that the seasonal distributions of NH$_3$ vary depending on the dominant source type and remains a very important element in understanding local emission sources and changing in environmental conditions (Tang et al., 2018b).

Soil emissions of NO$_x$ over north equatorial Africa (2.2 TgN/year) account for almost 70% of African soil emissions, because of the vast areas covered by dry ecosystems (Jaeglé et al., 2004). In the Sahel region, NH$_3$ emissions can represent an important

N flux in natural ecosystems, cropland, grazed soils (Hickman et al., 2018) and bacterial decomposition of urea in animal excreta (Adon et al., 2010). Indeed, many organisms in soils involved in the decomposition of organic matter excrete NH$_3$ directly or N compounds that readily hydrolyze to NH$_x$ (Bouwman et al., 1997). A minimum level of soil moisture is required for the microbial activities, such as urea hydrolysis, that generates NH$_3$ (Warner et al., 2017). Atmospheric NH$_3$ has been reported to be influenced by meteorological and physical parameters such as the presence of plants. Due to high temperatures,

low soil moisture and bare soil surfaces conditions, the process of volatilization from soils remains the dominant NH$_3$ loss in the West African Sahel region (Delon et al., 2010) and Africa contributes to 14% of the global source of NH$_3$ (Bouwman et

al., 1997). Likewise, NH$_3$ volatilization potential from soil/vegetation systems nearly doubles with every 5 °C increase in air temperature (Sutton et al., 2013; Pinder et al., 2012). However, the capture of NH$_3$ at the external surface of the leaf and transport into the leaf interior can be an important sink of atmospheric NH$_3$ (Van Hove et al., 1987).

According to Giglio et al. (2010), ~250 Mha of land area was burned in the Northern Hemisphere and Southern Hemisphere Africa for the time period 1997 through 2008. This value represents on average 70% of the global area burned each year. Biomass burning emits large amounts of aerosols and trace gases which significantly affect biosphere-atmosphere interface, atmospheric chemistry, cloud properties, Earth radiation budget, global carbon cycle, ecosystem and biodiversity, air quality and atmospheric circulation (Crutzen and Andreae, 1990; Andreae and Merlet, 2001; Stocker et al., 2013). Recently, Bray et

al. (2021) estimated average NH$_3$ emissions from biomass burning at a global scale over the period 2001-2015 at 4.53±0.51 Tg yr$^{-1}$.. Many scientific papers have shown that biomass burning represents the major source of NH$_3$ occurring in African savanna and forest ecosystems (Shi et al., 2015; van der Werf et al., 2017). The amount of NH$_3$ emitted from biomass burning in Africa represents roughly 60% to 70% of global NH$_3$ emissions biomass fires (Whitburn et al., 2015). Biomass burning emissions tend to drive seasonal variation in NH$_3$ total column densities in West Africa, with the largest emissions occurring

late in the dry season and early rainy season. Relationships between biomass burning and NH$_3$ may be observed when evaluating national scale statistics: countries with the highest rates of increasing Vertical Column Densities (VCDs) of NH$_3$ also had high rates of growth in CO VCDs; burned area displayed a similar pattern, though not significantly (Hickman et al., 2021).

Satellite measurements of NH$_3$ provide a means to monitor atmospheric composition globally (Clarisse et al., 2009; Warner et

al., 2017) and is a powerful tool for understanding atmospheric composition particularly for regions like Africa, where other types of measurements are scarce (Hickman et al., 2018). The Infrared Atmospheric Sounding Interferometer (IASI) datasets have been validated based on aircraft and ground-based measurements. The IASI version 3 NH$_3$ measurements are accurate at the scale of an individual pixel size of 12 km in diameter (Guo et al., 2021). Previous validation work comparing older versions of the IASI product with ground-based Fourier transform infrared (FTIR) observations of NH$_3$ total columns has also shown

robust correlations at sites with high NH$_3$ concentrations, but lower at sites where atmospheric concentrations approach IASI's detection limits (Dammers et al., 2017). Although FTIR observations are absent from Africa, earlier works have shown fair agreement between previous versions of IASI total column densities and INDAAF NH$_3$ surface observations in West Africa (Van Damme et al., 2015) and seasonal pattern (Hickman et al., 2018; Ossohou et al., 2019). During the year 2008, Hickman et al. (2018) found elevated total columns of NH$_3$ from the IASI in the Sahel during March-April mainly due to the Birch

effect. Through recent improvements in retrieval algorithms Van Damme et al. (2021) used the version 3 of the IASI-NH$_3$ total column datasets to characterize the evolution of atmospheric NH$_3$ at global, national and regional scales over the 11-year period (2008-2018). Using a statistical trend method based on least squares regression and bootstrap resampling, Van Damme et al.( 2021) found large increases of NH$_3$ in several subcontinental regions over the last decade, especially in western and central Africa (29.0 ± 2.3 % decade$^{-1}$).

Based on a 10-year period of ground-based measurements within the framework of the International Network to Study Deposition and Atmospheric Chemistry in Africa (INDAAF) program, Adon et al. (2010) documented surface concentrations and seasonal cycles according to the atmospheric sources of $NH_3$ in West and Central Africa. INDAAF has been a long-term monitoring measurement network since 1995 to document atmospheric chemistry and deposition fluxes in Africa. This program is part of the European Aerosol, Clouds and Trace Gases Research Infrastructure-France (ACTRIS-FR) and of the

International Global Atmospheric Chemistry / Deposition of Biogeochemically Important Trace Species (IGAC/DEBITS) activity. In addition, it is a labeled contributing network to the Global Atmospheric Watch/ World Meteorological Organization (GAW/WMO) program. INDAAF measurements are of great interest to remove some uncertainties in order to understand the seasonality of several trace gases including $NH_3$ in Western and Central Africa. Some uncertainties are caused by the scarcity of data on the spatial and temporal distribution of application of synthetic fertilizers and animal manure by crop, and the

prevailing management conditions (Beusen et al., 2008).

Here we provide updated analyses of these long-term records, complemented with satellite retrievals, to better understand 21[st] century $NH_3$ dynamics in Africa. Specifically, in the framework of the INDAAF program, this study aims to improve long-term $NH_3$ assessment in the Western and Central Africa region. We first compare the monthly and seasonal patterns in ground-based $NH_3$ concentrations (1998/2005-2018) and IASI $NH_3$ total columns (2008-2018) measured at three major African

ecosystems: dry savannas, wet savannas and forests. Monthly and seasonal evolutions allow us to highlight the main sources and factors influencing atmospheric $NH_3$ levels in the tropical African ecosystems. Secondly, we use non-parametric statistically robust tests to assess long-term trends of $NH_3$ from surface and satellite measurements over the ecosystemic transect, and discuss results according to the analysis of sources seasonality and meteorological data trends.

**2 Material and methods**

**2.1 Presentation of sampling sites**

Figure 1 shows the location of the 8 labeled INDAAF monitoring stations situated in West and Central Africa. Each site represents an African regional ecosystem with its own characteristics in terms of emission sources and its sensitivity to climatic, ecological, and anthropogenic changes. Thus, the sites are distributed by pairs according to latitudinal bands with significant

different rainfall patterns to represent dry savanna (Banizoumbou in Niger, Katibougou in Mali), wet savanna (Djougou in Benin, Lamto in Côte d'Ivoire) and equatorial forest (Bomassa in Republic of Congo, Zoétélé in Cameroon) ecosystems. Additional details on the monitoring sites can be found in the literature (Abbadie, 2006; Adon et al., 2010; Akpo et al., 2015; Delmas et al., 1995; Diawara et al., 2014; Le Roux et al., 2006; Ossohou et al., 2019; Ouafo-Leumbe et al., 2018; Yoboué et al., 2005). To date, measurements of atmospheric and meteorological physico-chemical parameters are continuing at all the

INDAAF sites. These measurements are referenced in the INDAAF database (http://indaaf.obs-mip.fr) and in the WMO OSCAR database (https://oscar.wmo.int/surface/#/).

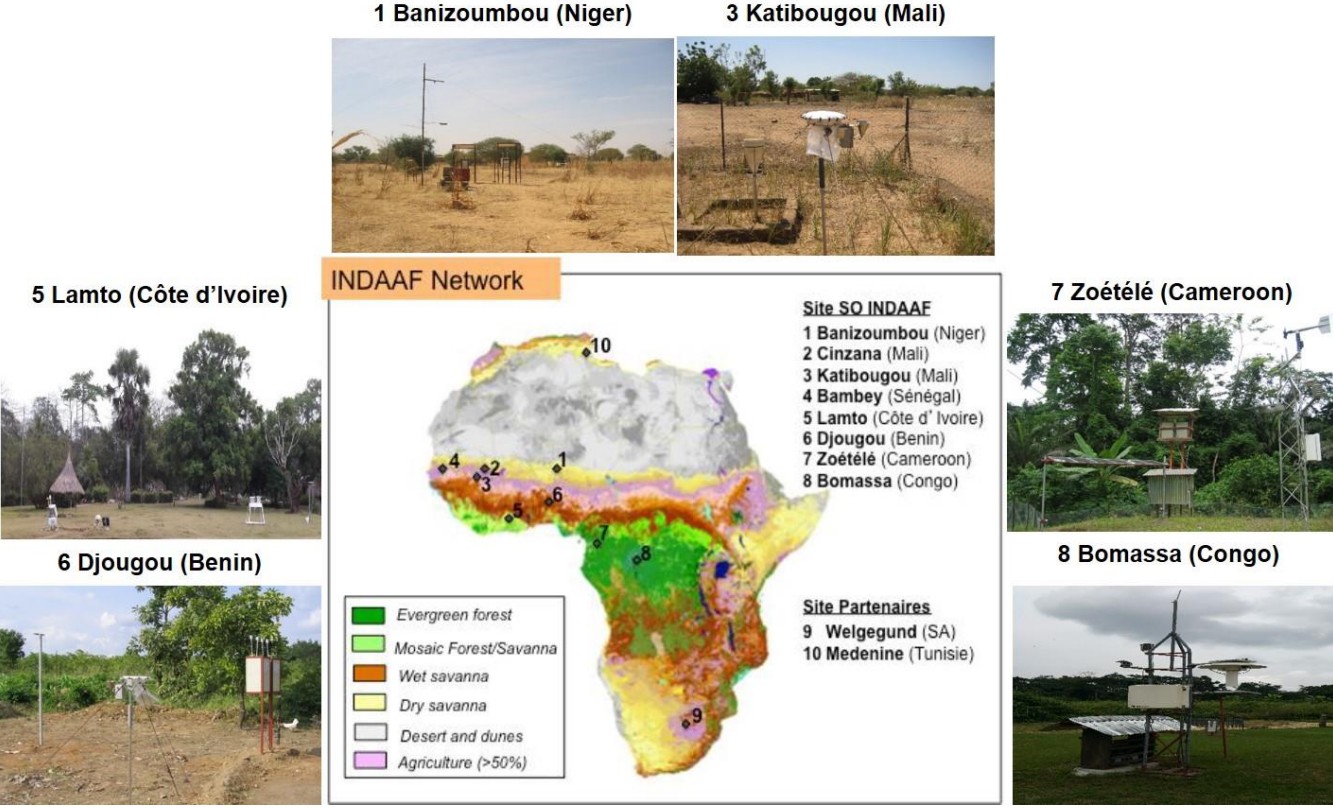

**Figure 1.** INDAAF measurement network composed by 10 stations across Africa. Presentation of the stations of (1) Banizoumbou (Niger), (3) Katibougou (Mali), (5) Lamto (Côte d'Ivoire), (6) Djougou (Benin), (7) Zoétélé (Cameroon) and (8) Bomassa (Republic of Congo) stations (Adapted from Mayaux et al. (2004); Ossohou et al. (2019)).

The geographical characteristics, soil, vegetation, climate types and the months representative of the wet and dry seasons of the western and Central African sites of interest are described in Table 1. It is important to keep in mind that dry savannas are characterized by a short wet season from June to September, whereas the wet season is longer in wet savanna and forest ecosystems extending from April to October and March to November, respectively.

**Table 1.** Site coordinates and location information (WS: wet season; DS: dry season). Dry savannas (WS: June–September DS: October–May), wet savannas (WS: April-October DS: November–March), forest (WS: March–November DS: December–February).

| Ecosystems | Station | Latitude, Longitude | Type of soil and/or vegetation | Climate | Country |
| --- | --- | --- | --- | --- | --- |

| Dry savannas | Banizoumbou (Adon et al., 2013; Delon et al., 2012; de Rouw and Rajot, 2004) | 13°31' N, 02°38' E | 91.2% Sandy soils, Tiger bush – fallow bush | Sahelian | Niger |
|---|---|---|---|---|---|
| | Katibougou (Adon et al., 2013; Delon et al., 2012) | 12°56' N, 07°32' W | Sandy soils, Deciduous shrubs | Sudano-Sahelian | Mali |
| Wet savannas | Djougou (Akpo et al., 2015; Ouafo-Leumbe et al., 2018) | 09°39' N, 01°44' E | Ferralitic and ferruginous soil, Mosaic of dry forests and savannah | Sudano-Guinean | Benin |
| | Lamto (Abbadie, 2006; Yoboué et al., 2005) | 06°13' N, 05°02' W | Ferrugineous soils, Grass, shrub and tree stratum | Guinean | Côte d'Ivoire |
| Forests | Bomassa (Mitani et al., 1993) | 02°12' N, 16°20' E | Dense evergreen forest | Equatorial | Republic of Congo |
| | Zoétélé (Sigha et al., 2003) | 03°15' N, 11°53' E | Dense evergreen forest | Equatorial | Cameroon |

**2.2 NH$_3$ sampling and chemical analysis**

Monitoring of NH$_3$ in the framework of the INDAAF program began in 1998 (2005 for Djougou). Sampling was carried out using the INDAAF passive sampler technique inspired by the work of Ferm (1991). The passive samplers were mounted and analyzed at the Laboratoire d'Aérologie (LAERO) in Toulouse (France) for all INDAAF sites.

Adon et al. (2010) give a complete overview of the sampling and analytical procedures for the INDAAF passive sampler
technique. For each INDAAF site, the passive samplers were made of impregnated filter paper with a species-specific solution for adsorption of gases. Samplers are exposed during one month in duplicates to ensure reproducibility and monthly concentrations are calculated from the arithmetic mean of the duplicates. Desorbed filters are analyzed using ion chromatography (IC). The Laboratoire d'Aérologie participates twice a year in WMO's quality assurance intercomparison program. Results have always shown that the analytical accuracy of the IC realized at the LAERO is greater than 95%. The
intercomparison results of the LAERO are available under the reference 700106 on the WMO website (http://qasac-

). The sampling technique using the INDAAF passive sampler method has been validated on tropical, subtropical, rural and urban sites in Africa (Adon et al., 2010; Bahino et al., 2018; Ossohou et al., 2020). INDAAF passive samplers have proven to be accurate, cheaper, easy to use and useful for air quality monitoring.

The precision of the measurements of passive samplers, evaluated through covariance with duplicates, was estimated at 14.3% for $NH_3$ (Adon et al., 2010). Detection limit for $NH_3$ was calculated from field blanks and is equal to 0.7±0.2 ppb. Values below the detection limit, as well as non-valid reproducibility values, were removed from the database. Thus, the percentages of valid data in the final database for the studied period 1998/2005-2018 were 97% for Banizoumbou, 93% for Katibougou, 90% for Djougou, 94% for Lamto, 73% for Bomassa and 93% for Zoétélé.

**2.3 Biomass burning and anthropogenic emissions of $NH_3$**

The fourth version of the Global Fire Emissions Database (GFED4) provides monthly biomass burning emissions at 0.25° resolution since 1997 from all biomass burning sources, i.e. many sectors (agricultural waste burning, boreal forest fires, peat fires, savanna fires, grassland fires, shrubland fires, temperate forest fires and tropical deforestation and degradation). Emissions of $NH_3$ from biomass burning sources were downloaded for the 1° x 1° grid cell containing each INDAAF site. The

GFED4 emissions are based on the combination of satellite information on fire activity and vegetation productivity to estimate gridded monthly burned area and fire emissions, as well as scalars that can be used to calculate higher temporal resolution emissions (Giglio et al., 2013; van der Werf et al., 2017). The Global Fire Emissions Database—currently by far the most widely used global fire emissions inventory—has been widely cited in the literature, and GFED4 data can be downloaded from the Emissions of atmospheric Compounds and Compilation of Ancillary Data (ECCAD) database (https://eccad3.sedoo.fr/).

The Community Emissions Data System (CEDS) produces consistent estimates of global air emissions species from anthropogenic sources (Smith et al., 2015; O'Rourke et al., 2021). The goal of the CEDS system is to combine existing emissions estimates with driver data to be able of producing consistent estimates of emissions over time at 0.1° x 0.1°. Here, we use CEDS anthropogenic emissions of $NH_3$ by all the sectors (Energy, transportation, ships, residential, industry process, solvents, agriculture and waste) at 1° x 1° grid cell containing each INDAAF site to estimate the anthropogenic $NH_3$ emissions.

Several studies have described $NH_3$ emissions data from CEDS (Hoesly et al., 2018; Feng et al., 2019; Beale et al., 2022).

**2.4 IASI $NH_3$ total columns and TRMM measurements**

IASI-A, launched aboard the European Space Agency's Metop-A in 2006, provides measurements of atmospheric $NH_3$ twice a day (9:30 in the morning and evening, Local Solar Time at the equator) (Clarisse et al., 2009). Here we use morning

observations, when the thermal contrast is more favorable for infrared retrievals in the lowest layers of the atmosphere (Clarisse et al., 2010; Van Damme et al., 2014). The $NH_3$ retrieval product used (ANNI-$NH_3$-v3R) follows a neural network retrieval approach. We refer to Whitburn et al. (2016) and Van Damme et al. (2017, 2021) for a detailed description of the algorithm.

Only observations with cloud cover below 10% were used. Given the absence of hourly or even daily observations of $NH_3$ concentrations in sub-Saharan Africa, the detection limit of IASI is difficult to determine with certainty. We used the Level-2 IASI-A $NH_3$ product observations within 100 km around each site for the years 2008 —the first full year of data available— to the end of 2018. It is important to note that monthly IASI $NH_3$ total columns can be negative. The negative total columns are related to measurement noise, inherent to any type of measurement. In the ANNI product, noise is translated both to positive and negative columns, unlike some other measurement products that translate noise always to positive columns, resulting in positive biases (Whitburn et al., 2016). On average, such noise averages out over long time periods, resulting in a column close to zero over remote regions. The few negative monthly values that are observed here, are close to zero, and occur during months with low ammonia and low measurement sensitivity. Note that these few negative values do not drive the observed trends.

We also used the Tropical Rainfall Measuring Mission (TRMM) daily precipitation product (3B42), which is based on a combination of TRMM observations, geo-synchronous infrared observations, and rain gauge observations (Huffman et al., 2007). Independent rain gauge observations from West Africa have been used to validate this precipitation product, with no indication of bias (Nicholson et al., 2003).

For analyses of seasonal and interannual variability in each product, we used the mean monthly value $NH_3$ or the monthly sum (for precipitation) for a 1° x 1° grid cell containing each INDAAF site. Note that monthly means are excluded from the IASI $NH_3$ analysis for months when there are fewer than 20 valid observations per month.

## 2.5 Trend analysis

Trend analyses were carried out by using Mann-Kendall (MK) (Kendall, 1975; Mann, 1945) and Seasonal Kendall (SK) (Hirsch et al., 1982) which are statistical non-parametric tests used to determine the increasing or decreasing trends of a random variable over some period of time. The MK and SK tests were suitable for cases with monotonic trends. The MK test allows working with no seasonal or other cycles in the data such as average annual data. The SK test follows the same principle as the MK test and is significantly robust to seasonality and was therefore applied for monthly time series. The SK test takes into account a 12-month seasonality in the time series data by computing the MK test on each dataset of "months" over the period 1998/2005-2018 separately, and then combining the results (Tang et al., 2018a). MK and SK tests allow working with non-normal data, in situations with many missing values, and are resistant to outliers (Kumar et al., 2018).

We coupled MK and SK tests respectively to Sen's Slope (SS) (Sen, 1968) and Seasonal Kendall Slope (SKS) to estimate the magnitude of the trend. These statistical tests have been widely applied and described in the literature to estimate trend in environmental parameters (Shadmani et al., 2012; Yue et al., 2002; Yue and Wang, 2004), while the application over African rural sites is limited (Ossohou et al., 2019, 2020). Two-tailed tests are conducted with the statistical software R version 4.0.4 (R Core Team, 2021) and (Addinsoft, 2022) for this study.

**3 Results and Discussion**

**3.1 Variations of ground-based NH₃ and IASI NH₃**

In the first part of this subsection, we will present the $NH_3$ concentration variations in the dry savanna ecosystem. In the second part, we will present the same variations for wet savanna and forest sites. Each part will show the monthly, annual evolutions and descriptive statistics of $NH_3$ ground-based and satellite measurements at each site. At the end of each of these two parts, we will discuss the results obtained according to the sources and the major processes that influence the atmospheric $NH_3$ levels.

**3.1.1 Dry savanna**

Figure 2 presents monthly variations of ground-based $NH_3$ concentrations and IASI $NH_3$ at the INDAAF dry savanna sites of Banizoumbou (Figure 2a) and Katibougou (Figure 2b) over 1998-2018 and 2008-2018, respectively. The monthly 21-year surface concentrations of $NH_3$ are in the same range at Banizoumbou and Katibougou (Table 2) with coefficients of variation of ~50%. Nethertheless, the monthly coefficient of variation of IASI $NH_3$ total columns appear to be larger at Banizoumbou (52%) compared to Katibougou (43%) over the 11-year period. From 2008 to 2018, we obtain a significant Pearson's correlation between monthly ground-based $NH_3$ concentrations and IASI $NH_3$ total columns at Banizoumbou (*r=0.30, p<0.01*), but not at Katibougou (*r=0.06, p=0.51*).

Table 2 shows that mean ground-based concentrations of $NH_3$ for each dry savanna site are significantly higher in wet season compared to dry season according to the t-test *(p<0.01)*. Furthermore, we observe a decrease of 8.1% (in Banizoumbou) and 23.0% (in Katibougou) in average annual ground-based concentrations of $NH_3$ from the period 1998-2007 to 2008-2018 (Table 3). In contrast, Table 3 shows that average wet season ground-based $NH_3$ concentrations increase between 1998-2007 and 2008-2018 in Banizoumbou (+5.9%), but decrease in Katibougou (-22.0%). Mean dry season $NH_3$ ground-based concentrations decrease in Banizoumbou (-18.9) and Katibougou (-22.0) from 1998-2007 to 2008-2018 (Table 3).

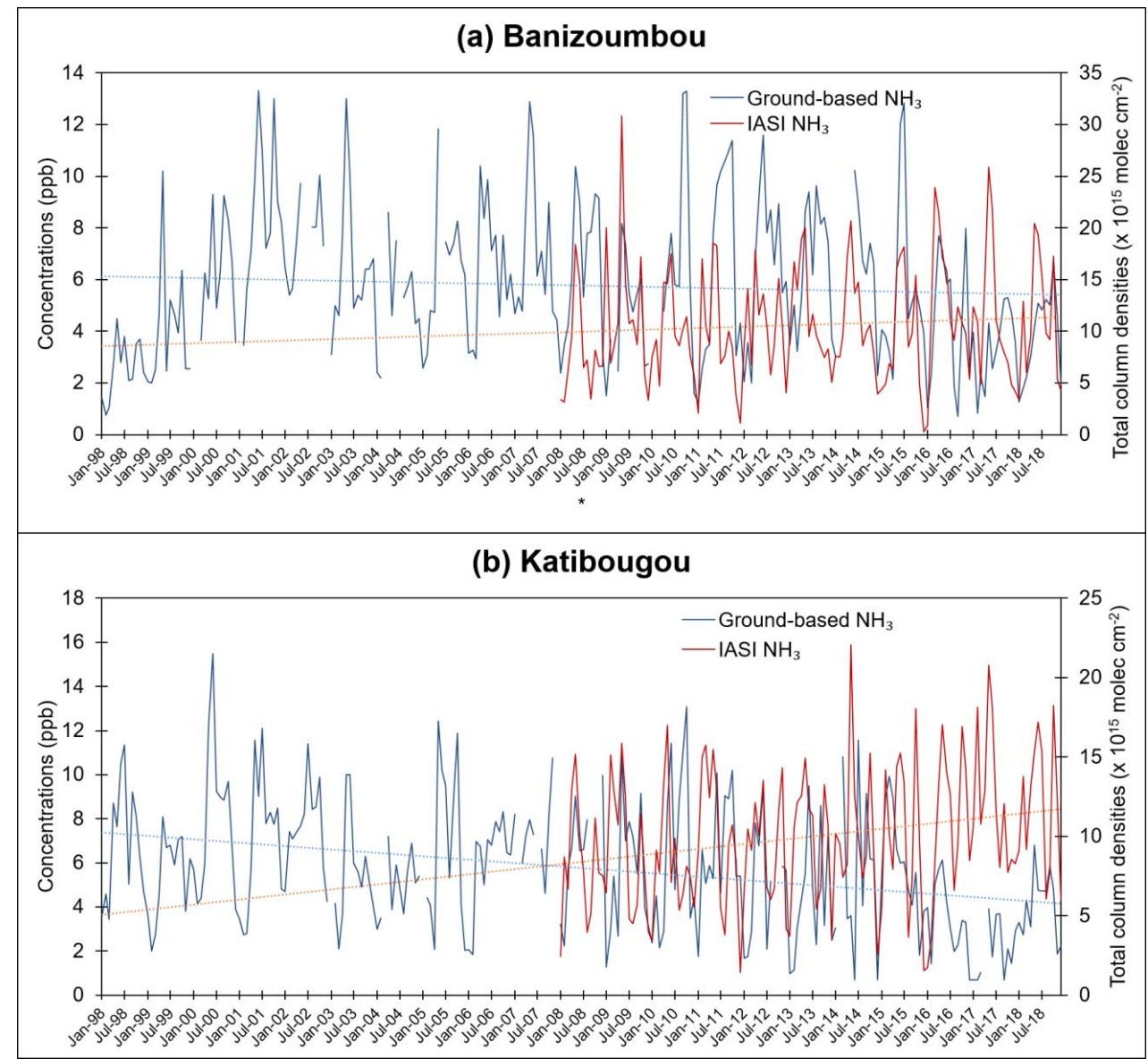

**Figure 2.** Monthly time-series of ground-based NH₃ concentrations over the period 1998–2018, IASI NH₃ total column densities from 2008 to 2018 at (a) Banizoumbou, Niger and (b) Katibougou, Mali. The dashed lines represent the linear regression lines.

**Table 2.** Minimum (Min), maximum (Max) and average (Avg) monthly, annual and seasonal INDAAF NH₃ ground-based concentrations (1998–2018), and IASI NH₃ total column densities (2008–2018) at Banizoumbou, Niger and Katibougou, Mali

| | | Ground-based NH₃ (ppb) | IASI NH₃ ($10^{15}$ molec cm⁻²) |
|---|---|---|---|
| | | | |

|  |  | Banizoumbou | Katibougou | Banizoumbou | Katibougou |
|---|---|---|---|---|---|
| Monthly | Min | 0.7 | 0.7 | 0.3 | 1.4 |
|  | Max | 13.3 | 13.1 | 30.8 | 22.1 |
| Annual | Avg | 5.8±1.2 | 5.7±1.1 | 10.7±1.1 | 10.0±0.9 |
| Wet Season | Min | 2.7 | 2.4 | 8.0 | 6.8 |
|  | Max | 10.4 | 9.3 | 13.5 | 12.4 |
|  | Avg | 6.9±1.6 | 6.7±1.5 | 11.3±1.3 | 9.1±1.6 |
| Dry Season | Min | 2.5 | 1.8 | 7.4 | 9.0 |
|  | Max | 8.1 | 8.0 | 12.9 | 12.2 |
|  | Avg | 5.2±1.1 | 5.2±1.0 | 10.3±1.5 | 10.3±0.8 |

**Table 3**. Minimum (Min), maximum (Max) and average (Avg) monthly, annual and seasonal ground-based $NH_3$ concentrations in ppb (1998-2007 & 2008-2018) at Banizoumbou, Niger and Katibougou, Mali

|  |  | 1998-2007 | | 2008-2018 | |
|---|---|---|---|---|---|
|  |  | Banizoumbou | Katibougou | Banizoumbou | Katibougou |
| Monthly | Min | 0.8 | 1.8 | 0.7 | 0.7 |
|  | Max | 13.3 | 13.1 | 13.3 | 13.1 |
| Annual | Avg | 6.1±1.3 | 6.5±0.8 | 5.6±1.1 | 5.0±1.1 |
| Wet Season | Min | 2.7 | 5.0 | 3.6 | 2.4 |
|  | Max | 9.8 | 9.3 | 10.4 | 9.0 |
|  | Avg | 6.7±1.5 | 7.7±1.3 | 7.1±1.8 | 5.9±1.5 |
| Dry Season | Min | 2.5 | 4.8 | 3.2 | 1.8 |

| | Max | 8.1 | 8.0 | 6.0 | 6.4 |
| --- | --- | --- | --- | --- | --- |
| | Avg | 5.8±1.2 | 5.9±0.7 | 4.7±0.8 | 4.6±1.0 |

250

Figure 3 compiles the mean monthly 21-year average ground-based concentrations (gray line) and 11-year average total column densities (dark line) in dry savanna ecosystems to obtain the mean annual cycle evolutions of $NH_3$ at the stations of Banizoumbou (a) and Katibougou (b). In both sites of the sub Saharan dry ecosystems, we observe a marked seasonal cycle with two peaks in ground-based concentrations and total columns of $NH_3$ appearing at the beginning (May-June) and the end

(October) of the wet season (Figure 3). The lowest values of $NH_3$ (concentrations and densities) are generally observed during December-January, but the highest values are obtained in May-June. Indeed, mean monthly measurements vary at Banizoumbou from 2.8±1.1 ppb (January) to 8.3±2.6 ppb (June) for ground-based concentrations, and from 4.0±1.3 x $10^{15}$ molec cm$^{-2}$ (January) to 19.5±3.7 x $10^{15}$ molec cm$^{-2}$ (May) for IASI $NH_3$ total columns (Figure 3a). At Katibougou, mean monthly ground-based concentrations ranged from 3.3±1.3 ppb (January) to 7.6±2.1 ppb (June), and from 4.8±2.1 x $10^{15}$ molec

260 cm$^{-2}$ (December) to 16.2±2.2 x $10^{15}$ molec cm$^{-2}$ (May) for IASI $NH_3$ total columns (Figure 3b). The mean annual variation coefficients are 34% and 27% for ground-based concentrations, 40% and 34% for IASI $NH_3$ total column measurements at Banizoumbou and Katibougou, respectively.

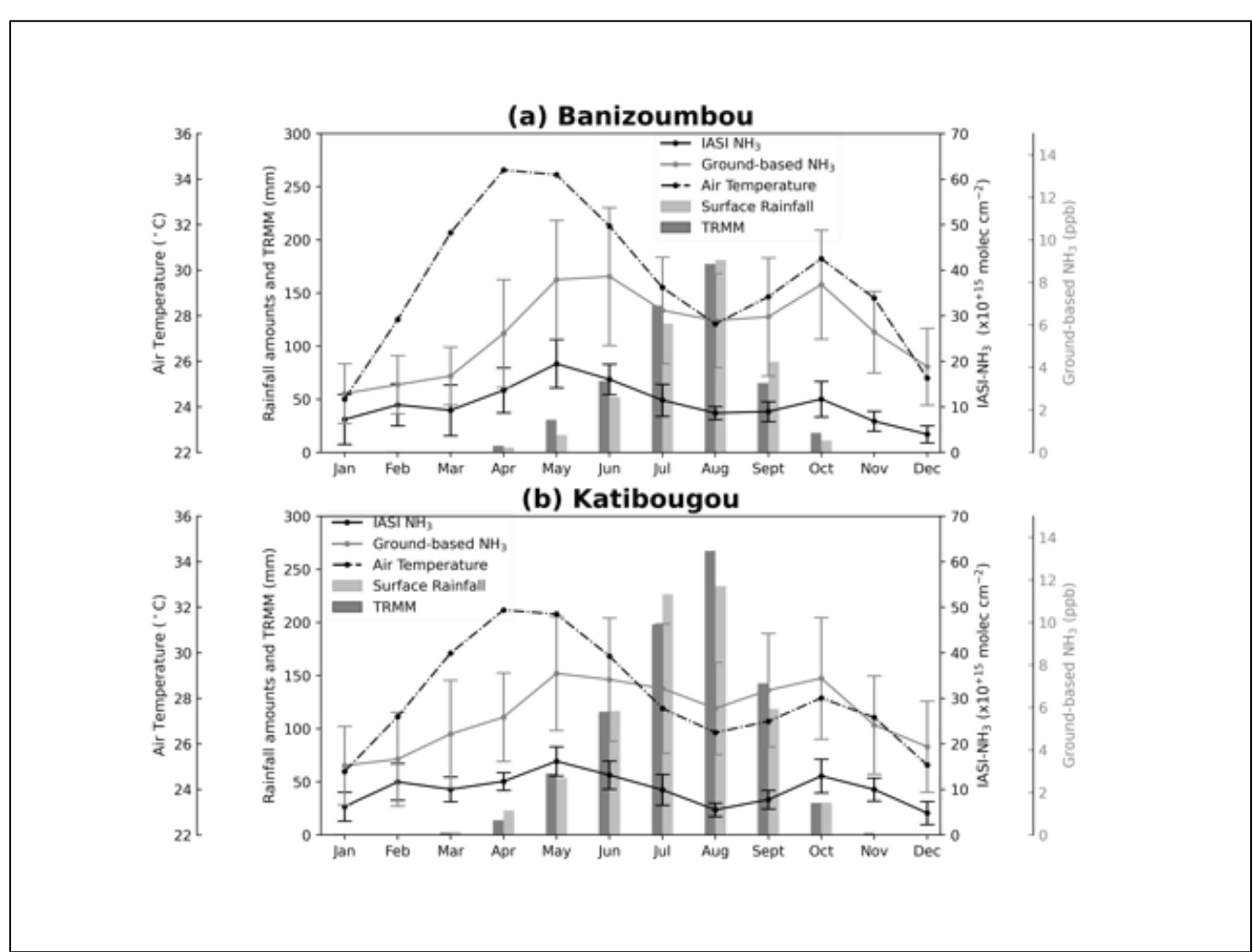

**Figure 3.** Mean monthly ground-based NH₃ concentrations (1998–2018), IASI NH₃ total column densities (2008–2018), rainfall amounts, air temperatures (1998-2018) measured by ground-based instruments (1998-2018) and TRMM (2005-2018) at (a) Banizoumbou, Niger and (b) Katibougou, Mali. Error bars represent the mean absolute deviation.

In the Sahelian region, major sources of atmospheric NH₃ include bacterial decomposition of urea in livestock excreta and emission from natural or fertilized soils (Bouwman and Van Der Hoek, 1997). In addition, it has been shown in the literature that African dry savanna ecosystems, characterized by sandy soils, tiger bush, fallow bush and deciduous shrubs (Ossohou et al., 2019), tend to have alkaline soils, creating favorable conditions to NH₃ volatilization (Clarisse et al., 2019; Delon et al., 2017; Hickman et al., 2018; Vågen et al., 2016). Among other factors, air temperature, rainfall pattern and amount influence NH₃ emissions considerably in drylands like Banizoumbou and Katibougou. We find positive correlation coefficients statistically significant at 99% between ground-based NH₃ concentrations and air temperature in the dry savanna ecosystem of Banizoumbou (0.32) and Katibougou (0.26) showing that NH₃ volatilization increases with temperature (Van Damme et al.,

2020). Note that the correlation coefficient between monthly IASI $NH_3$ and emission fluxes $NH_3$ from agricultural sector from CEDS emission inventory at Katibougou is equal to 0.22 *(p<0.05)*. In the dry savannas, soils are often characterized by large pulses of $NH_3$ related to successive dryings and rewettings of dry soils (McCalley and Sparks, 2008; Soper et al., 2016). As we can see in Figure 3, the first peak observed in May-June (beginning of the wet season) could be related to the optimal soil moisture to initiate bacterial activity and a flush of newly mineralized N. Our results support the conclusions of an earlier study that used satellite retrievals and in situ measurements for the year 2008 over Africa to argue that the onset of the rainy season causes pulsed emissions of $NH_3$ over the Sahel (Hickman et al., 2018). Our study based on ground-based and satellite measurements ranging from one to two decades clearly shows a correspondence between early rainy season precipitation and $NH_3$ concentrations over the two dry savanna sites. Moreover, the results based on our analysis of a long-term database clearly indicates that this process is reproducible every year.

The temporal evolution of $NH_3$ can be associated with two most important phenomena: (1) Possible Birch effect emissions in the early and possibly late rainy season, and (2) the overall seasonal cycle of $NH_3$ and the reasons for this broad seasonality (separate from the Birch effect). Indeed, during the wet season (June-September), the months are wetter and cooler and give the soils more moisture than the dry season months. As a result, wet season soils are less susceptible to intense drying events than during the dry season. This consequently results in more limited $NH_3$ volatilization from soil drying, leading to low $NH_3$ levels in the wet season. However, at the end of the wet season, rainfall became erratic and led to drying soils for a few days. This erratic rainfall combined with an increase in air temperature may explain the second observed $NH_3$ peak, occurring at the end of the wet season. A similar late-season pulse of nitric oxide from the re-wetted soils was observed at the regional scales in the Sahel (Jaeglé et al., 2004), suggesting that there may be some similar potential for $NH_3$ pulsing from re-wetted dry soils. This late-season peak appears to be of less importance than the early wet season peak, presumably because over the growing season, growing vegetation, and microbial communities that immobilized and reduced soils nitrogen pools and may continue to do so. One of the arguments for why Birch effect emissions happen at the beginning of the growing season is that there has been an accumulation of labile N in soils in the dry season. During the wet season, $NH_3$ is found directly in the rainwater in the form of $NH_4^+$, thus promoting wet deposition on the growing vegetation. $NH_3$ also react with some acid gases such as $H_2SO_4$, $HNO_3$ and $HCl$ to form aerosols of atmospheric ammonium salts, such as ammonium sulphate ($[NH_4]_2SO_4$), ammonium bisulphate ($NH_4HSO_4$), ammonium nitrate ($NH_4NO_3$) and ammonium chloride ($NH_4Cl$). The conversion of gases to particles in the atmosphere can occur through condensation and/or direct nucleation processes (Baek et al., 2004). Condensation adds mass to pre-existing aerosols, while direct nucleation allows the formation of atmospheric aerosols from gaseous precursors. These reactions could therefore lead to a decrease in atmospheric $NH_3$ concentrations in the Sahelian region (Koziel et al., 2006). As shown by positive correlation between IASI $NH_3$ and emissions of $NH_3$ by agricultural activities, we suggest that in the dry savanna area of Katibougou, $NH_3$ concentration levels could also be influenced by agriculture activities. Since the months of September through March correspond to the fire period in the wet savanna and forest sites, we suggest that even though the two dry savanna sites experience few fires, $NH_3$ columns from IASI are certainly affected by $NH_3$ present in the transported fire plumes. It's also important to highlight the pastoralism in the Sahel region, mainly

nomadic in nature. Indeed, Sahelian agro-pastoralism appears to be very important, representing 25 to 30% of the Gross Production Product (GDP), and contributes to 10 to 15% of the GDP of Mali and Niger for example (Adon et al., 2010).

### 3.1.2 Wet savanna and forest

In the wet savanna ecosystem, we present the monthly evolutions of ground-based $NH_3$ concentrations (2005-2018: Djougou,
1998-2018: Lamto) and IASI $NH_3$ total column densities (2008-2018 for both sites) at Djougou (Figure 4a) and Lamto (Figure 4b). Monthly ground-based $NH_3$ concentrations range from 0.7 to 12.1 ppb at Djougou and from 0.7 to 14.2 ppb at Lamto. IASI $NH_3$ total column densities vary from -2.8 x $10^{15}$ to 36.6 x $10^{15}$ molec cm$^{-2}$ at Djougou and from -1.3 x $10^{15}$ to 58.0 x $10^{15}$ molec cm$^{-2}$ at Lamto. The results show that the maxima of IASI $NH_3$ total column densities are highest at Lamto (Figure 4). The coefficients of variation are globally high, equal to 57% and 62% for ground-based measurements, and 54% and 69%
for IASI $NH_3$ total columns at Djougou and Lamto, respectively. For the entire period of measurements, Pearson correlation test applied to monthly ground-based $NH_3$ concentrations and IASI $NH_3$ total columns reveals no significant correlation at Djougou (*r=0.03, p=0.72*), but strong linear correlation at Lamto (*r=0.55, p<0.01*).

Table 4 presents a synthesis of monthly, seasonal and annual minimum, maximum and average ground-based $NH_3$ concentrations and IASI $NH_3$ total columns at Djougou and Lamto stations. The results show that mean annual, wet season
and dry season ground-based $NH_3$ concentrations in Djougou are significantly higher than that in Lamto (t-test, *p<0.05*). In contrast, mean annual and dry season IASI $NH_3$ total columns are significantly higher (*t-test, p<0.01*) at Lamto compared to Djougou (Table 4). The data recorded in Table 5 show that for the period 2006-2007, the average ground-based $NH_3$ concentrations at Djougou and Lamto are of the same order of magnitude. However, the average concentration obtained in the dry season of the period 2006-2007 in Lamto is 20% higher than in Djougou. In contrast to the period 2006-2007, annual, dry
and wet season averages of ground-based concentrations of $NH_3$ over the period 2008-2018 are the highest at Djougou, with differences ranging from 1.6 to 1.8 ppb compared to the Lamto averages (Table 5).

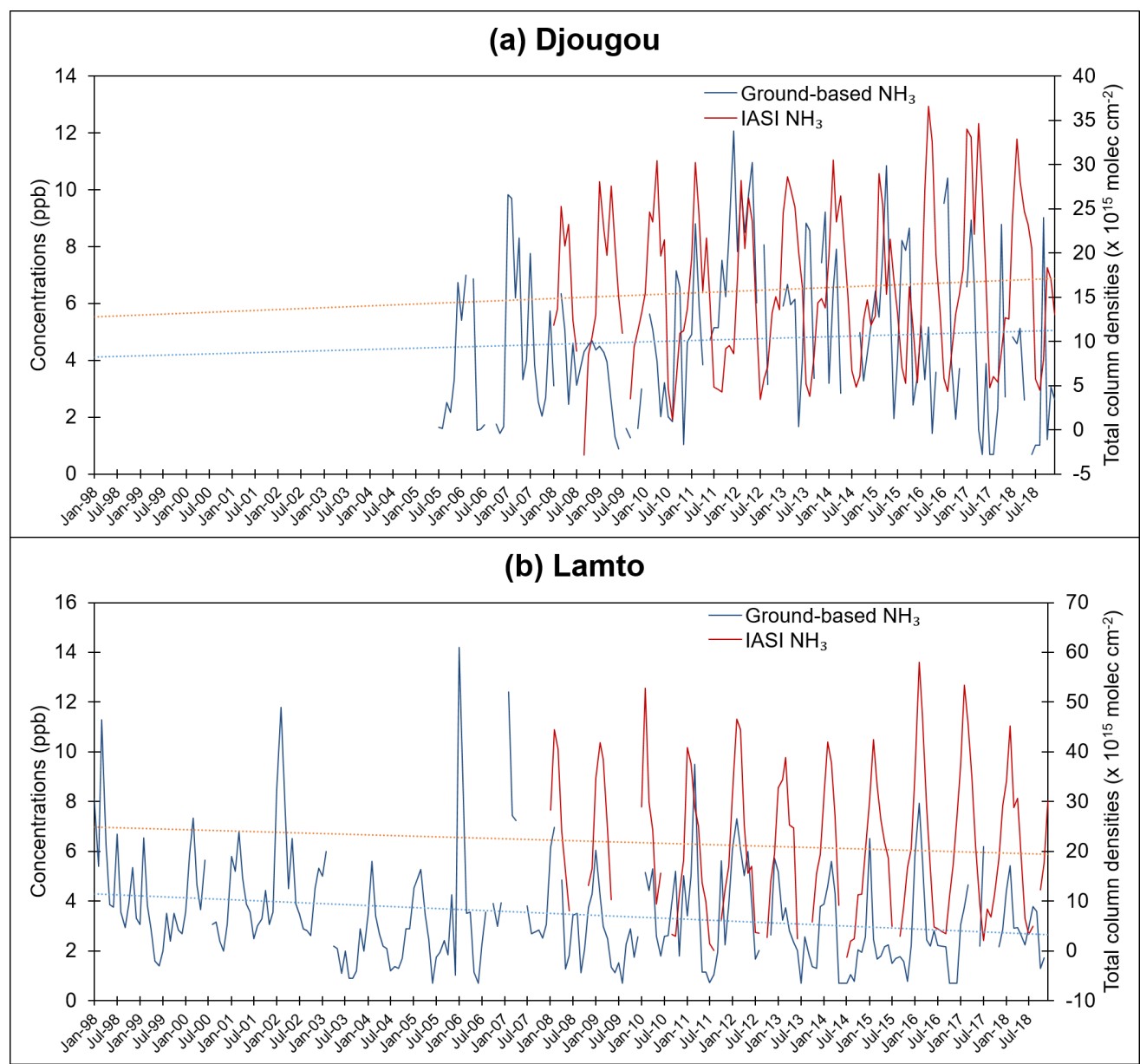

**Figure 4.** Monthly time-series of ground-based NH$_3$ concentrations over the periods 2005-2018 and 1998–2018, IASI NH$_3$ total column densities from 2008 to 2018 at (a) Djougou, Benin and (b) Lamto, Côte d'Ivoire. The dashed lines represent the linear regression lines.

**Table 4.** Minimum (Min), maximum (Max) and average (Avg) monthly, annual and seasonal INDAAF NH$_3$ ground-based concentrations (Djougou : 2005-2018; Lamto : 1998–2018), and IASI NH$_3$ total column densities (2008–2018) at Djougou, Benin and Lamto, Côte d'Ivoire

| | | Ground-based NH$_3$ | IASI NH$_3$ |
|---|---|---|---|
| | | | |

| | | (*ppb*) | | (*10<sup>15</sup> molec cm<sup>-2</sup>*) | |
|---|---|---|---|---|---|
| | | **Djougou** | **Lamto** | **Djougou** | **Lamto** |
| Monthly | Min | 0.7 | 0.7 | -2.8 | -1.3 |
| | Max | 12.1 | 14.2 | 36.6 | 58.0 |
| Annual | Avg | 4.7±1.3 | 3.5±0.8 | 16.0±1.3 | 20.9±1.6 |
| Wet Season | Min | 1.5 | 1.5 | 10.6 | 8.4 |
| | Max | 7.6 | 4.5 | 15.0 | 15.9 |
| | Avg | 4.1±1.5 | 2.8±0.9 | 13.4±0.9 | 12.0±1.7 |
| Dry Season | Min | 3.5 | 2.7 | 14.9 | 27.3 |
| | Max | 8.8 | 7.8 | 23.1 | 37.1 |
| | Avg | 5.5±1.3 | 4.6±1.1 | 19.6±2.1 | 30.7±2.7 |

**Table 5**. Minimum (Min), maximum (Max) and average (Avg) monthly, annual and seasonal ground-based $NH_3$
concentrations in ppb (2006-2007 & 2008-2018) at Djougou, Benin and Lamto, Côte d'Ivoire. *Full year of data in Djougou begins in 2006.*

| | | **2006-2007**[*] | | **2008-2018** | |
|---|---|---|---|---|---|
| | | **Djougou** | **Lamto** | **Djougou** | **Lamto** |
| Monthly | Min | 1.4 | 0.7 | 0.7 | 0.7 |
| | Max | 9.8 | 14.2 | 12.1 | 9.5 |
| Annual | Avg | 4.4±1.1 | 4.7±0.3 | 4.9±1.3 | 3.2±0.6 |
| Wet Season | Min | 2.7 | 3.5 | 1.5 | 1.5 |
| | Max | 4.6 | 3.9 | 7.6 | 4.5 |

| | Avg | 3.6±0.9 | 3.2±0.7 | 4.4±1.5 | 2.6±0.7 |
|---|---|---|---|---|---|
| Dry Season | Min | 3.9 | 6.4 | 3.5 | 2.7 |
| | Max | 6.8 | 6.7 | 8.8 | 5.6 |
| | Avg | 5.4±1.5 | 6.5±0.2 | 5.6±1.4 | 4.0±0.7 |

Figure 5 presents the annual mean cycle of monthly ground-based concentrations and IASI $NH_3$ total column densities at Djougou (Figure 5a) and Lamto (Figure 5b) located in the wet savanna ecosystem. The results show that the annual mean ground-based and IASI $NH_3$ profiles have a poor covariation at Djougou (Figure 5a), while IASI $NH_3$ shows a good agreement with ground-based $NH_3$ at the Lamto site (Figure 5b). Ground-based $NH_3$ concentrations and IASI $NH_3$ total columns exhibit a seasonality at Lamto and Djougou stations with higher values occurring in the dry season (January to March) and lower values in the wet season (May through November). Mean annual cycle of IASI $NH_3$ total column densities seasonality are less marked at Djougou compared to Lamto station. Monthly mean concentrations and total column densities of $NH_3$ range from 3.2±1.4 (June) to 6.7±1.8 ppb (February) and from 4.7±1.1 x $10^{15}$ (August) to 27.3±3.9 x $10^{15}$ molec $cm^{-2}$ (February) at Djougou (Figure 5a), and from 2.2±1.0 (June) to 6.1±1.8 ppb (February) and from 2.7±1.6 x $10^{15}$ (July) to 45.3±5.3 x $10^{15}$ molec $cm^{-2}$ (February) at Lamto (Figure 5b), respectively. The mean annual variation coefficients are 23% and 41% for ground-based concentrations, 51% and 76% for IASI $NH_3$ total column measurements at Djougou and Lamto, respectively.

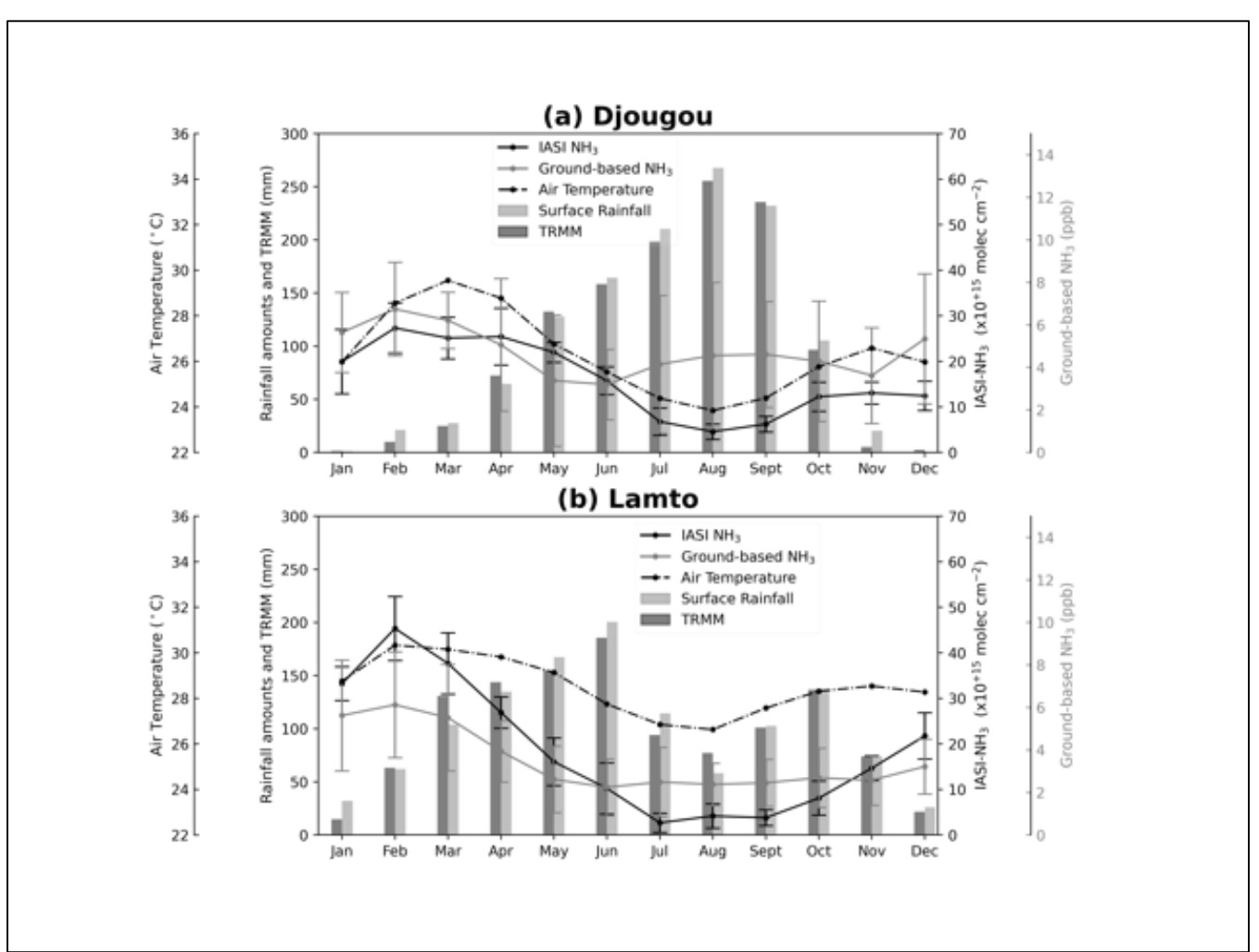

**Figure 5.** Mean monthly ground-based NH₃ concentrations (Djougou : 2005-2018 & Lamto : 1998–2018), IASI NH₃ total column densities (2008–2018), rainfall amounts, air temperatures measured by ground-based instruments (1998-2018) and TRMM (2005-2018) (Djougou : 2005-2018 & Lamto : 1998–2018) at (a) Djougou, Benin and (b) Lamto, Côte d'Ivoire. Error bars represent the mean absolute deviation.

The monthly variations of ground-based NH₃ concentrations (1998-2018) and IASI NH₃ total column densities (2008-2018) over the two forested monitoring sites are presented in Figure 6. The results show that the peak values of ground-based concentrations and IASI NH₃ total column densities are generally higher at Bomassa (Figure 6a) compared to Zoétélé (Figure 6b). The monthly 21-year coefficients of variation of NH₃ are in the same order of magnitude at Bomassa (55%) and Zoétélé (56%). Nethertheless, the monthly coefficient of variation of IASI NH₃ total column densities are significantly higher in the forested ecosystem compared to dry and wet savannas, i.e more than 80% at Bomassa and Zoétélé over the 11-year period. Significant Pearson's correlations are found between monthly ground-based NH₃ and IASI-NH₃ total column densities at Bomassa ($r = 0.18, p = 0.07$) and Zoétélé ($r = 0.34, p < 0.01$).

The monthly, seasonal and annual measurement results of ground-based $NH_3$ concentrations and IASI $NH_3$ total columns at Bomassa and Zoétélé are summarized in Table 6. According to the t-test, ground-based $NH_3$ average concentrations are significantly higher ($p<0.001$) at Bomassa compared to Zoétélé, but IASI $NH_3$ total column average densities are in the same order of magnitude for these sites. For each forested ecosystem station, the results show that the mean ground-based $NH_3$ concentrations are in the same order of magnitude between wet and dry seasons. However, IASI total column densities are significantly higher (*t-test, p<0.001*) in the dry season at Bomassa and Zoétélé compared to the wet season (Table 6). We summarized the descriptive statistics of ground-based $NH_3$ concentrations in Table 7, separately for the periods 1998-2007 and 2008-2018. In general, we note that mean concentrations have increased for the Bomassa site, but decreased at Zoétélé between 1998-2007 and 2008-2018. Indeed, mean dry season, annual and wet season ground-based $NH_3$ concentrations in Bomassa have increased by 33, 41, and 44%, respectively, from 1998-2007 to 2008-2018 (Table 7). At Zoétélé, we observe a decrease of about 26% in the mean annual, wet and dry seasons ground-based concentrations of $NH_3$ between 1998-2007 and 2008-2018 (Table 7).

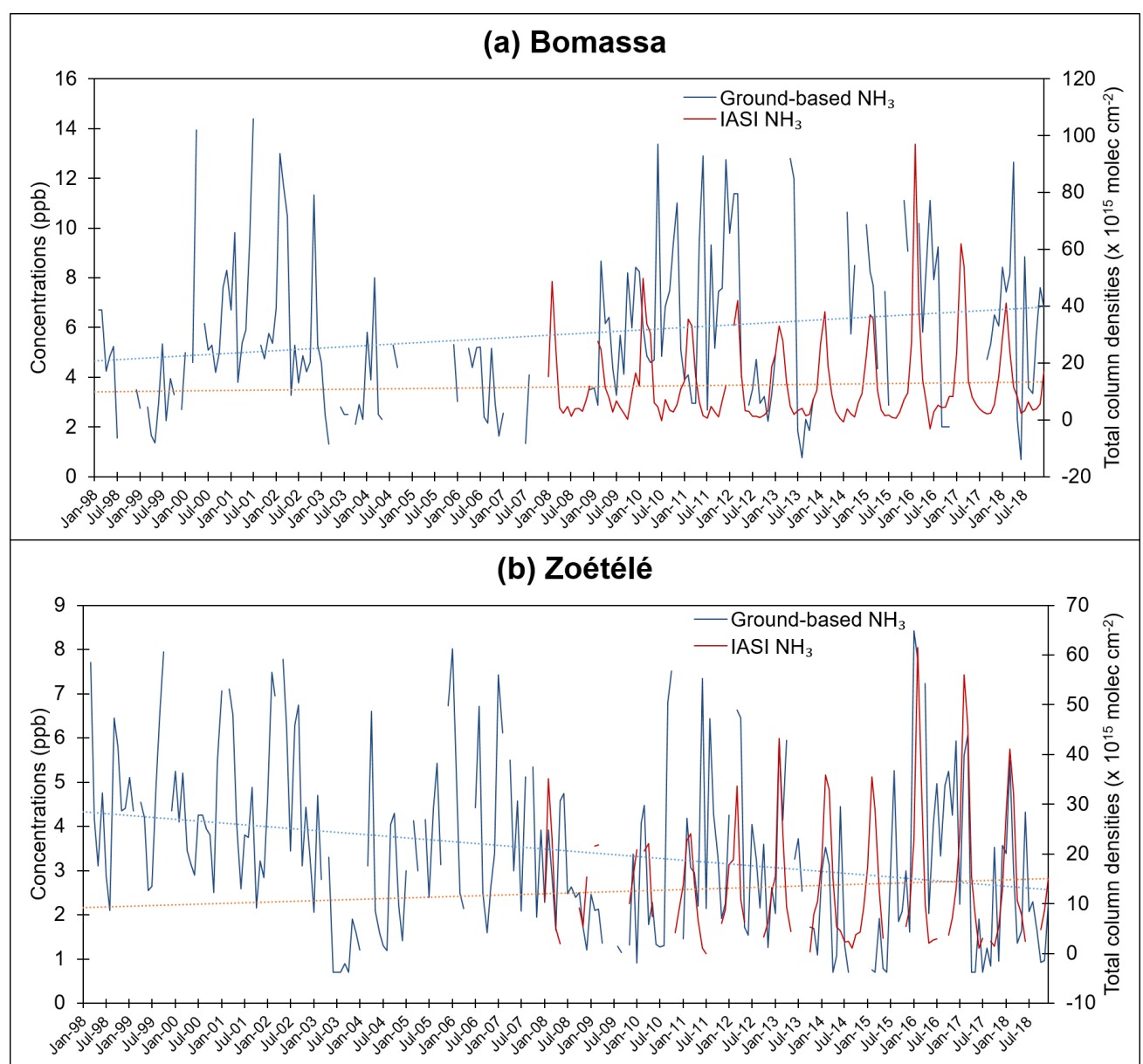

**Figure 6.** Monthly time-series of ground-based NH₃ concentrations over the period 1998–2018, IASI NH₃ total column densities from 2008 to 2018 at (a) Bomassa, Republic of Congo and (b) Zoétélé, Cameroon. The dashed lines represent the linear regression lines.

**Table 6.** Minimum (Min), maximum (Max) and average (Avg) monthly, annual and seasonal INDAAF NH₃ ground-based concentrations (1998–2018), and IASI NH₃ total column densities (2008–2018) at Bomassa, Republic of Congo and Zoétélé, Cameroon.

| | | Ground-based NH₃ (*ppb*) | IASI NH₃ (*$10^{15}$ molec cm$^{-2}$*) |
|---|---|---|---|
| | | | |

|  |  | Bomassa | Zoétélé | Bomassa | Zoétélé |
|---|---|---|---|---|---|
| Monthly | Min | 0.7 | 0.7 | -3.1 | -0.1 |
|  | Max | 14.4 | 8.4 | 97 | 61.5 |
| Annual | Avg | 5.6±1.4 | 3.4±0.9 | 12.4±2.1 | 13.8±2.0 |
| Wet Season | Min | 2.4 | 1.4 | 5.0 | 7.8 |
|  | Max | 8.6 | 5.6 | 10.9 | 12.4 |
|  | Avg | 5.6±1.5 | 3.3±0.9 | 7.9±1.2 | 9.9±1.4 |
| Dry Season | Min | 2.3 | 1.2 | 20.4 | 14.8 |
|  | Max | 9.1 | 7.4 | 44.1 | 32.5 |
|  | Avg | 5.6±1.7 | 3.8±1.3 | 26.8±4.8 | 22.0±4.8 |

**Table 7**. Minimum (Min), maximum (Max) and average (Avg) monthly, annual and seasonal ground-based $NH_3$ concentrations in ppb (1998-2007 & 2008-2018) at Bomassa, Republic of Congo and Zoétélé, Cameroon

|  |  | 1998-2007 |  | 2008-2018 |  |
|---|---|---|---|---|---|
|  |  | Bomassa | Zoétélé | Bomassa | Zoétélé |
| Monthly | Min | 1.3 | 0.7 | 0.7 | 0.7 |
|  | Max | 14.4 | 8.0 | 13.6 | 8.4 |
| Annual | Avg | 4.6±1.4 | 4.0±0.7 | 6.5±0.9 | 2.9±0.7 |
| Wet Season | Min | 2.4 | 1.5 | 4.4 | 1.4 |
|  | Max | 6.8 | 5.6 | 6.8 | 4.5 |
|  | Avg | 4.5±1.4 | 3.8±0.8 | 6.5±0.9 | 2.8±0.8 |
| Dry Season | Min | 2.3 | 1.3 | 3.5 | 1.2 |

| | | | | | |
|---|---|---|---|---|---|
| | Max | 8.4 | 5.8 | 9.1 | 7.4 |
| | Avg | 4.8±1.7 | 4.4±1.0 | 6.4±1.4 | 3.3±0.9 |

We present the mean annual ground-based $NH_3$ concentrations and IASI $NH_3$ evolutions based on monthly data measured in
the forested ecosystems of Bomassa (Figure 7a) and Zoétélé (Figure 7b). Ground-based concentrations show no clear
seasonality at Bomassa and Zoétélé. In contrast, IASI $NH_3$ total columns show a well-marked seasonality, with high densities
in the dry season (December to February), and low densities in the wet season (April to November) for the two sites. Mean
monthly ground-based $NH_3$ concentrations narrowly vary from a minimum of 4.1±1.1 ppb (September) and a maximum of
7.1±3.0 ppb (March) at Bomassa (Figure 7a), from a minimum and maximum of 2.4±0.9 ppb (November) and 4.2±1.6 ppb
(March) at Zoétélé, respectively (Figure 7b). $NH_3$ total column densities show a peak representing the annual maximum in
February (45.7±13.6 x $10^{15}$ molec $cm^{-2}$ for Bomassa and 37.1±9.9 x $10^{15}$ molec $cm^{-2}$ for Zoétélé) and the lowest values in July
(2.2±1.7 x $10^{15}$ molec $cm^{-2}$ for Bomassa and 2.1±1.0 x $10^{15}$ molec $cm^{-2}$ for Zoétélé). Mean annual coefficients of variation
over the 21-year period are 16% and 19% for $NH_3$ concentrations, and more than 80% for IASI $NH_3$ over the 11-year period
at Bomassa and Zoétélé, respectively.

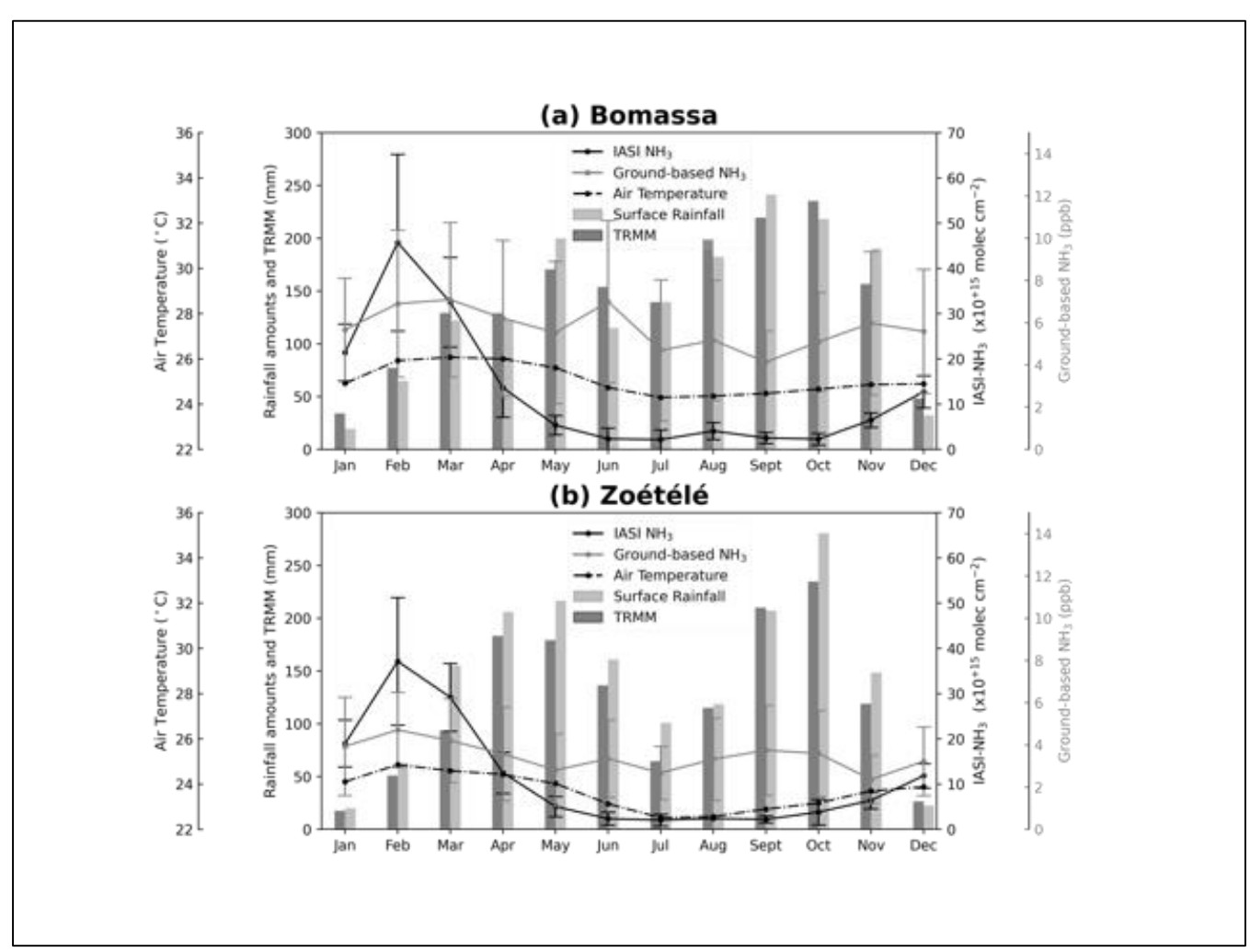

**Figure 7.** Mean monthly ground-based NH$_3$ concentrations (1998–2018), IASI NH$_3$ total column densities (2008–2018), rainfall amounts, air temperatures (1998-2018) measured by ground-based instruments (1998-2018) and TRMM (2005-2018) at (a) Bomassa, Republic of Congo and (b) Zoétélé, Cameroon. Error bars represent the mean absolute deviation.

Biomass burning is recognized as a significant source of atmospheric NH$_3$, especially in tropical regions, but also at higher latitudes (Coheur et al., 2009; Lutsch et al., 2019; Whitburn et al., 2015). It represents the second largest terrestrial source of NH$_3$ after agriculture (Whitburn et al., 2017) and contributes to about 13% of total NH$_3$ emissions (Galloway et al., 2004) at the global scale. Other major sources of NH$_3$ in African wet savanna and forest ecosystems include decomposition of urea from animal excreta, fertilized soils (Bouwman et al., 2002b) and domestic fuelwood burning (Adon et al., 2010; Lobert et al., 1990). In wet savannas and forests in Africa, the NH$_3$ concentrations represent a combination of all natural sources with the largest contribution from biomass burning sources (Adon et al., 2010).

Our study demonstrates that highest NH$_3$ concentrations in wet savanna and forest ecosystems are recorded during the period when fires predominate (December-February), while the lowest are obtained when rainfall is high. Indeed, during the dry season, farmers take advantage of the absence of rainfall to clear land, weed and burn agricultural residues. This slash-and-burn agriculture contributes significantly to nitrogen (NO$_x$ and NH$_3$) and carbon (CO and CO$_2$) emissions (Tiemoko et al., 2021) into the atmosphere during the dry season. Fires related to agriculture and hunting become more important in the dry season and represent respectively 64% and 6% of the economic activities of the villagers in certain areas such as Lamto (Suzanne, 2016).

In order to show the influence of the combustion and anthropogenic sources on the atmospheric NH$_3$ concentrations and total column densities, we have conducted linear correlation study between monthly ground-based NH$_3$ concentrations and IASI NH$_3$ total column densities on the one hand, and the GFED4 (1998-2018) and CEDS (1998-2018) emission data of NH$_3$ on the other hand. For the station of Djougou, correlations are calculated over the periods 2005-2018 and 2008-2018 for ground-based NH$_3$ and IASI NH$_3$, respectively. The Pearson correlation results are summarized in Table 8 below.

**Table 8**. Pearson correlation coefficients between NH$_3$ measurements (ground-based and IASI) and emission data (GFED4 and CEDS) at INDAAF wet savanna (Djougou, Benin and Lamto, Côte d'Ivoire) and forest (Bomassa, Republic of Congo and Zoétélé, Cameroon) sites over the period 1998-2018 (Djougou: 2005-2018). Coefficients have been calculated from monthly data. Values in brackets represent significance levels (*p-values*).

| | **Djougou** | **Lamto** | **Bomassa** | **Zoétélé** |
|---|---|---|---|---|
| **Ground-based and GFED4** | 0.04 *(0.63)* | **0.34** *(<0.01)* | -0.06 *(0.39)* | 0.12 *(0.07)* |
| **Ground-based and CEDS** | **0.19** *(0.02)* | -0.04 *(0.5)* | **0.22** *(<0.01)* | **-0.27** *(<0.01)* |
| **IASI and GFED4** | -0.06 *(0.5)* | **0.33** *(<0.01)* | **0.18** *(0.04)* | **0.39** *(<0.01)* |
| **IASI and CEDS** | **0.27** *(<0.01)* | **0.37** *(<0.01)* | **0.24** *(<0.01)* | **0.27** *(<0.01)* |

These results show that NH$_3$ emissions from biomass burning  influence ground-based concentrations of NH$_3$ in Lamto, and total column densities of NH$_3$ in the Lamto, Bomassa and Zoétélé areas (Table 8). Similarly, there is an evident relationship between anthropogenic emissions by residential and agricultural sectors in the wet savanna and forest sites (Table 8). These results are consistent with the study of Whitburn et al. (2015) carried out in four regions including "Africa north of Equator (ANE)" accounting for a major part of the total affected by fires. Indeed, they found a significant correlation (*r = 0.57*) between time series of monthly NH$_3$ columns retrieved from IASI measurements and  MODIS fire radiative power (FRP) over the period 2008-2013 (Whitburn et al., 2015).The most likely explanation of these significant correlations between NH$_3$ (simultaneously for ground-based concentrations and total columns) and emission data (GFED and CEDS) is that NH$_3$ concentrations observed at Lamto are mainly influenced by biomass burning and agricultural sources (Adon et al., 2010, 2013;

Whitburn et al., 2015). However, atmospheric $NH_3$ concentrations and columns in Djougou, Bomassa and Zoétéle are also affected by human activities in these areas.

For the wet savanna and forested ecosystems where $NH_3$ seasonality is driven by biomass burning emissions, it looks like there is still an overall pattern of increasing $NH_3$ in the dry season, and decreasing $NH_3$ in the rainy season that would be expected, which is not the case at Djougou. This modest increase in ground-based $NH_3$ concentrations in the wet season at Djougou

could be due to the Leaf Area Index (LAI) which is much lower there than in Lamto during the wet season with annual averages of about 1.2 $m^2$ $m^{-2}$ in Djougou and 3.6 $m^2$ $m^{-2}$ in Lamto (Ossohou et al., 2019). Indeed, $NH_3$ emissions during the wet season at Djougou are therefore less intercepted by the canopy via the dry deposition process. Note that canopy heights at Djougou and Lamto look very different on Figure 1. Dry deposition can be affected by both LAI and the vertical distance between canopy and instrument. The canopy looks much shorter at Djougou, with lots of vegetation being lower than the instrument.

This could be another reason why $NH_3$ emissions may be less intercepted by the canopy at Djougou. In a general way, we assume that canopy interception/bi-directional exchange could play a role in reducing the seasonal variability at the surface (Adon et al., 2013; Delon et al., 2019), but not for the total column densities while keeping in mind that the satellite observations are for 100 km around each site, so they are influenced by a lot of non-local dynamics. It is also important to highlight the influence of air temperature, which significantly enhance $NH_3$ volatilization at Djougou, Lamto and Bomassa

with correlation coefficients (*p<0.05*). The correlation coefficients will be given in section 3.2 to justify the significant $NH_3$ trends obtained for these sites.

## 3.2 Trends of ground-based $NH_3$ and IASI $NH_3$

We conduct the long-term trend computations by using Mann-Kendall (MK) test coupled to Sen Slope (SS) for mean annual,

mean wet and dry seasons for each year of ground-based concentrations (14 and 21-year periods) and total columns densities within a diameter of 100 km centered around each site (11-year period). Additional trend analyses are carried out using the Seasonal Kendall (SK) coupled to Seasonal Kendall Slope (SKS) only for monthly data over the entire period. We adopt significance thresholds of 90% (*p<0.1*) for all trend analyses, and the percent increase or decrease is based on the mean concentrations or total column densities over each period.

In section 3.2.1, we present and discuss trends results for mean annual, wet and dry seasons of ground-based concentrations and IASI $NH_3$ total column densities in the three main ecosystems using MK test coupled to SS. The section 3.2.2 focuses on long term trends based on monthly data of $NH_3$ ground-based concentrations and total column densities at the six stations by using the SK test coupled to SKS. In these sections, we present only the results of significant trends. In the paragraph preceding the conclusion of the paper, we present a general comment on the trends obtained for each ecosystem and explain the

differences obtained between ground-based concentration and total column density measurement trends.

Reported ground-based $NH_3$ concentration and IASI $NH_3$ trends are analyzed in the light of $NH_3$ emissions from biomass burning and anthropic sources (described in section 2.3), meteorological (air temperature and rainfall) and physical (LAI) parameters when available, which influence the atmospheric level of $NH_3$.

**3.2.1 Annual trends**

Globally, results indicate decreasing annual, wet and dry season trends in ground-based $NH_3$ concentrations for the three ecosystems except at Bomassa, but increasing trends in IASI $NH_3$ total column densities. At the annual scale, results show there is no simultaneous trend for ground-based concentrations and total column densities of $NH_3$ at the same site.

Results indicate significant increases in IASI $NH_3$ total column densities at the dry savanna of Katibougou site of +0.34 x $10^{15}$

molec $cm^{-2}$ $yr^{-1}$ (+3.46% $yr^{-1}$), at the wet savanna of Djougou site of +0.42 x $10^{15}$ molec $cm^{-2}$ $yr^{-1}$ (+2.65 % $yr^{-1}$) and at the forest of Zoétélé of +0.47 x $10^{15}$ molec $cm^{-2}$ $yr^{-1}$ (+3.42% $yr^{-1}$) over the 11-year period. Surprisingly, for the forested ecosystem, annual ground-based $NH_3$ concentrations register an increasing trend at Bomassa of +0.14 ppb $yr^{-1}$ (+2.56 % $yr^{-1}$) but a decreasing trend at Zoétélé of -0.10 ppb $yr^{-1}$ (-2.95 % $yr^{-1}$) over 21-year period.

We also investigate potential trends by applying the non-parametric MK test coupled to SS to the annual average of wet and

dry seasons (separately) at the six stations representing the great ecosystems in Sub Saharan Africa. We observe in the wet season that $NH_3$ concentrations decrease at Katibougou in Malian dry savanna by -0.22 ppb $yr^{-1}$ (-3.25% $yr^{-1}$), and at Zoétélé in Cameroon's forest ecosystem by -0.11 ppb $yr^{-1}$ (-3.24 % $yr^{-1}$) but increase at the other forested site of Bomassa in Republic of Congo by +0.13 ppb $yr^{-1}$ (+2.29 % $yr^{-1}$). Ground-based $NH_3$ concentrations in the dry season reveal decreasing trends in both dry savanna (-0.13 ppb $yr^{-1}$ or -2.41 % $yr^{-1}$ for Banizoumbou and -0.12 ppb $yr^{-1}$ or -2.26 % $yr^{-1}$ for Katibougou) and wet

savanna (-0.08 ppb $yr^{-1}$ or -1.70 % $yr^{-1}$ for Lamto) sites. From satellite measurements, the significant increasing trends are obtained from mean wet season-to-mean wet season for IASI $NH_3$ total column densities at Banizoumbou (+0.36 x $10^{15}$ molec $cm^{-2}$ $yr^{-1}$ or 3.20% $yr^{-1}$), Katibougou (+0.55 x $10^{15}$ molec $cm^{-2}$ $yr^{-1}$ or 6.01% $yr^{-1}$) and Djougou (+0.24 x $10^{15}$ molec $cm^{-2}$ $yr^{-1}$ or 1.77% $yr^{-1}$). In the dry season, we obtain increasing trends at Katibougou (+0.24 x $10^{15}$ molec $cm^{-2}$ $yr^{-1}$ or 2.33% $yr^{-1}$), Djougou (+0.69 x $10^{15}$ molec $cm^{-2}$ $yr^{-1}$ or 3.54% $yr^{-1}$) and Zoétélé (+1.37 x $10^{15}$ molec $cm^{-2}$ $yr^{-1}$ or 6.24% $yr^{-1}$).

To investigate the potential causes of the observed trends of $NH_3$ concentrations at Zoétélé, we have applied MK trend and Pearson's correlation tests to meteorological and $NH_3$ emission data from GFED4 databases. The results show the decreasing trend in ground-based $NH_3$ concentrations in the wet season at Zoétélé could be attributed to wet season-to-wet season increasing of the LAI (+0.69% $yr^{-1}$), with a 99% significant anticorrelation of -0.57 between these two variables. We do not yet know the cause of the increase in LAI from one wet season to the next in the Zoétélé forest ecosystem. However, more

vegetation results in greater dry deposition rate, which would significantly reduce the observed wet season to wet season ground-based atmospheric $NH_3$ concentrations at Zoétélé. During the wet season, air humidity and soil moisture increase, leading to large $NH_3$ deposition on vegetation during wet months (Delon et al., 2019).

The increasing trends in IASI NH$_3$ total column densities at Katibougou (dry and wet seasons) and Djougou (annual, dry and wet seasons) are linked to emissions from the agricultural sector. Indeed, our results show significant correlations at 95%
between satellite data and NH$_3$ emissions from agriculture at Katibougou ($\tau = 0.75$ and 0.96 in dry and wet seasons, respectively) and Djougou ($\tau = 0.72$, 0.63 and 0.68 for annual, dry season and wet seasons, respectively). In addition, MK test coupled to SS applied to agricultural NH$_3$ emissions provided by the CEDS database shows increasing trends for annual means (+2.51% yr$^{-1}$ at Djougou), annual average of wet season (+2.36% yr$^{-1}$ at Katibougou and +2.56% yr$^{-1}$ at Djougou) and annual average of dry season (+2.37% yr$^{-1}$ at Katibougou and +2.47% yr$^{-1}$ at Djougou) over the 2008-2018 period.

### 3.2.2 Trends accounting for seasonality

Long time series of atmospheric NH$_3$ could usually be affected by seasonality, which is the cyclical changes in concentrations or densities over the course of the year. The SK test is significantly robust in revealing trends in seasonal time series. In this section, we perform trend computations using SK coupled to SKS of monthly mean ground-based NH$_3$ concentrations and
IASI NH$_3$ total column densities of the entire dataset. Results for only significant monthly trends (p<0.1) from all INDAAF sites are shown in Figure 8. In general, the statistical tests reveal significant decreasing trends for ground-based NH$_3$ concentrations (except at Bomassa), but increasing trends for IASI NH$_3$ total column densities (Figure 8).

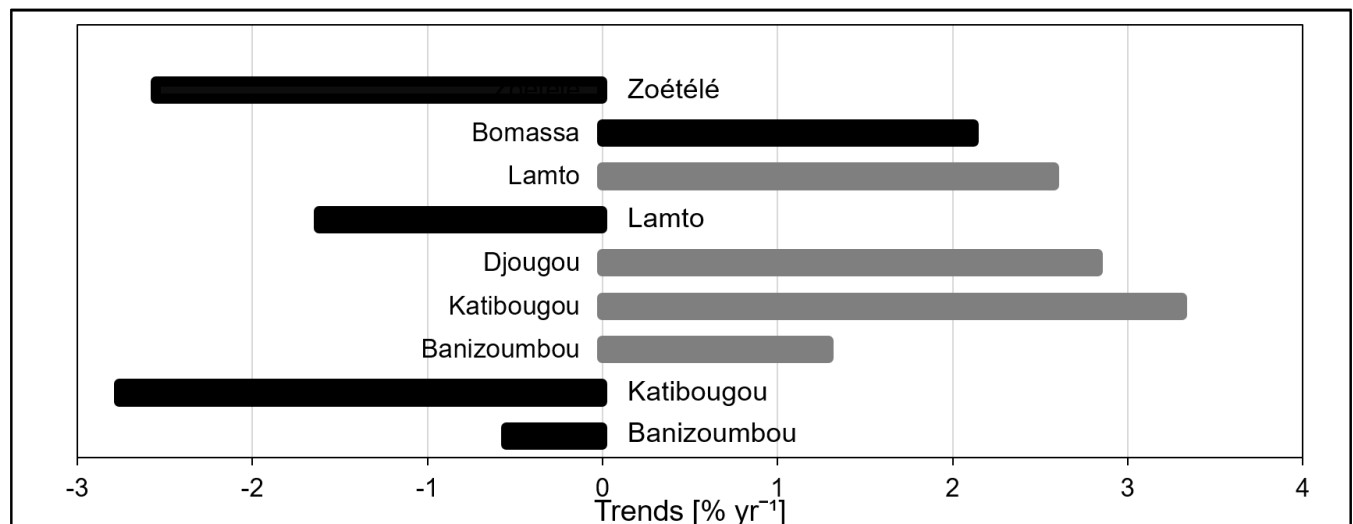

**Figure 8.** Estimated percentage changes in ground-based NH$_3$ concentrations (in black) and IASI NH$_3$ total column densities (in grey) for
INDAAF network stations where monthly trends are significant at 90%. These trends are obtained using the monthly data of the overall database.

Ground-based NH$_3$ concentrations decrease in the dry savannas of Banizoumbou by -0.03 ppb yr$^{-1}$ (-0.55 % yr$^{-1}$) and Katibougou by -0.16 ppb yr$^{-1}$ (-2.76 % yr$^{-1}$), but IASI NH$_3$ total column densities increase at the same sites by +0.14 x 10$^{15}$
molec cm$^{-2}$ yr$^{-1}$ (+1.29 % yr$^{-1}$) and +0.33 x 10$^{15}$ molec cm$^{-2}$ yr$^{-1}$ (+3.31 % yr$^{-1}$), respectively. A significant decreasing trend is

also found for ground-based $NH_3$ concentrations in the wet savanna of Lamto (-0.06 ppb $yr^{-1}$ or -1.62 % $yr^{-1}$), but significant increasing trends are obtained for IASI $NH_3$ total column densities both at Djougou (+0.45 x $10^{15}$ molec $cm^{-2}$ $yr^{-1}$ or +2.83 % $yr^{-1}$) and Lamto (+0.54 x $10^{15}$ molec $cm^{-2}$ $yr^{-1}$ or +2.58 % $yr^{-1}$) sites. SK test applied to monthly ground-based $NH_3$ concentrations in the forested ecosystem sites shows significant increasing trend by +0.12 ppb $yr^{-1}$ (+2.12 % $yr^{-1}$) at Bomassa, but decreasing trend by -0.09 ppb $yr^{-1}$ (-2.55 % $yr^{-1}$) at Zoétélé. On the contrary, we obtained no significant trends for IASI $NH_3$ total column densities in the forested zoneThe SK test applied to monthly data from January, 1998 to December, 2018 shows that relative humidity decreases by -0.15% $yr^{-1}$ at Lamto. We calculate Pearson's correlation between ground-based $NH_3$ concentrations and relative humidity and we find a coefficient of -0.50 significant at 99%. This statistical test demonstrates that the decreasing monthly trend of ground-based $NH_3$ concentrations cannot be explained by the monthly relative humidity trend.

In the dry savanna ecosystem, we believe that the simplest explanation for the increasing trend of IASI $NH_3$ total column densities observed at Katibougou is the significant increasing trend of $NH_3$ emissions by agricultural activities (+2.33% $yr^{-1}$) for the period 2008-2018. In the wet savanna ecosystems, we suggest that the increase in agricultural emissions of $NH_3$ (+2.02% $yr^{-1}$) and air temperature (+0.06°C $yr^{-1}$) for the period 2008-2018 at Djougou could be responsible for the increasing trend in IASI $NH_3$ total column densities at this site, with 99% significant correlation coefficients of 0.36 and 0.65, respectively. At Lamto, the increase in total column densities of $NH_3$ could be attributed to increases in air temperature (+0.14°C $yr^{-1}$) and emissions of $NH_3$ by agricultural activities (+2.3% $yr^{-1}$) over the period 2008-2018. Indeed, there are significant linear correlations between IASI $NH_3$ and air temperature ($\tau = 0.72$, $p<0.01$), and between IASI $NH_3$ and agricultural emissions of $NH_3$ ($\tau = 0.50$, $p<0.01$). At the forested ecosystem of Bomassa, our results show that monthly air temperature and $NH_3$ emissions by anthropogenic sources (residential and agriculture) contribute to the increasing of ground-based concentrations and total column densities of $NH_3$. Indeed, during ground-based concentrations and IASI total column densities observations, we find increasing trends of air temperature (1998-2018 : +0.03°C $yr^{-1}$, 2008-2018 : +0.02°C $yr^{-1}$), $NH_3$ emissions from residential (1998-2018 : +5.78% $yr^{-1}$, 2008-2018 : +4.74% $yr^{-1}$) and agriculture (1998-2018 : +4.46% $yr^{-1}$, 2008-2018 : +1.52% $yr^{-1}$) sectors at Bomassa. In addition, our study shows that monthly air temperature and these anthropogenic sources of $NH_3$ are significantly correlated with ground-based and IASI $NH_3$ at Bomassa over these two periods of data. We assume that increasing trends of atmospheric $NH_3$ at Bomassa are due to air temperature, residential and agricultural $NH_3$ emissions.

Trend studies of $NH_3$ concentrations and densities obtained respectively with the INDAAF passive samplers and the IASI instrument have shown significant trends depending on each biome. Overall, we obtained decreasing trends for ground-based

measurements (except at the Bomassa forest site), but increasing trends for IASI total column densities of $NH_3$ (except in forest

ecosystems). The long-term statistical trend results for $NH_3$ emissions from GFED4 database are not significant, so biomass burning could not explain the trends obtained for the ground-based and satellite data. However, meteorological and anthropogenic emission data from CEDS clearly show that the drivers of atmospheric $NH_3$ trends in the (1) dry savanna of Katibougou is agriculture, (2) wet savanna of Djougou and Lamto are air temperature and agriculture, and (3) forest of Bomassa are air temperature, residential and agriculture.


In our study, satellite observations integrated across 100 km centered around each site can be expected due to the very different nature of the observations. IASI provides a total column value, which we have averaged over an area of roughly 7,854 square km for each station comparison. The surface stations provide a point measurement at the surface. So any differences between a) surface concentrations and concentrations at any other altitude in the atmosphere or b) between composition at the station

and at any other point in the 100 square-mile area can produce a mis-match between the station observations and the IASI retrieval. In addition, $NH_3$ plumes from the combustion of biomass from distant sources are likely less well captured from INDAAF passive samplers, while they are very well measured by IASI (Zheng et al., 2021). It is likely that IR sounders have a higher sensitivity to fire plumes, which are located higher in the atmosphere (and so the ground-based measurements will show less sensitivity to them).


## 4 Conclusion

Using a 21-year period of INDAAF passive samplers and an 11-year period of IASI product, we have characterized coevolutions and trends of atmospheric $NH_3$ at six stations of the INDAAF network in the African dry savanna (Banizoumbou, Niger and Katibougou, Mali), wet savanna (Djougou, Benin and Lamto, Côte d'Ivoire) and forest (Bomassa, Republic of

Congo and Zoétélé, Cameroon). The remote sensing data is intended as a complement to the surface data, to provide some insight into local dynamics near each surface station. The results showed that ground-based concentrations of $NH_3$ and IASI $NH_3$ total column densities are significantly higher in the dry savanna and wet savanna ecosystems, respectively. Indeed, mean annual ground-based concentrations of $NH_3$ over periods 1998/2005-2018 period are 5.7-5.8 ppb in dry savanna, 3.5-4.7 ppb in wet savanna and 3.4-5.6 ppb in forest ecosystems. The overall mean annual IASI $NH_3$ total column densities for a circle

with a diameter of 100 km centered on each site over 2008-2018 are 10.1-11.0 x $10^{15}$ molec cm$^{-2}$ in the dry savanna, 16.5-21.4 x $10^{15}$ molec cm$^{-2}$ in the wet savanna and 14.3-15.1 x $10^{15}$ molec cm$^{-2}$ in the forest ecosystems. If we consider only ground-based measurements, the results show that $NH_3$ emissions from Sahelian soils, livestock and agriculture (only at Katibougou) in the dry savanna ecosystem lead to average concentrations in the dry season that are equal to those obtained in the wet savanna ecosystem, globally dominated by biomass burning and agriculture.

We have recorded 95% significant Pearson correlation between monthly ground-based concentrations and IASI total column densities of $NH_3$ at Banizoumbou (*r=0.30*), Lamto (*r=0.55*), Bomassa (*r=0.18*) and Zoétélé (*r=0.34*), showing that $NH_3$ abundancies at the wet savanna of Lamto show the best agreement between ground-based and satellite remote sensing. In the dry savanna sites of Banizoumbou and Katibougou, the seasonal ground-based concentrations of $NH_3$ are highest both at the beginning and the end of the wet season. Conversely, ground-based concentrations of $NH_3$ are highest in the dry season at the wet savanna sites of Djougou and Lamto, but no marked seasonality between wet and dry season was observed for ground-based $NH_3$ concentrations in the forest sites of Bomassa and Zoétélé. IASI $NH_3$ total column densities follow the same seasonality as ground-based $NH_3$ concentrations in the dry and wet savannas, while the seasonality is more marked in the forested ecosystem.

The non-parametric Mann-Kendall statistical trend test shows 90% significant mean annual increasing trend for IASI $NH_3$ total column densities which is the most important in the dry savanna of Katibougou ($+3.98 \% \ yr^{-1}$). Ground-based $NH_3$ concentrations in the forested ecosystem increase at Bomassa ($+2.56 \% \ yr^{-1}$), but decrease at Zoétélé ($-2.95 \% \ yr^{-1}$). In both dry and wet seasons, ground-based $NH_3$ concentrations decrease from $-3.25\% \ yr^{-1}$ (Katibougou) to $-1.70\% \ yr^{-1}$ (Lamto), but increase in wet season at Bomassa ($+2.29\% \ yr^{-1}$). IASI $NH_3$ total column densities increase in the wet season ($+6.66\% \ yr^{-1}$) and dry season ($+2.55\% \ yr^{-1}$) only at Katibougou, Mali. The seasonal Kendall test applied to monthly data over the entire periods also shows decreasing trends at all the sites, except at Bomassa ($+2.12\% \ yr^{-1}$) for ground-based $NH_3$ concentrations. In contrast to trends calculated using ground-based observations, monthly IASI $NH_3$ total column densities increase for all ecosystems, ranging from $+1.21\% \ yr^{-1}$ (Bomassa) to $+4.00\% \ yr^{-1}$ (Katibougou). The increasing trends observed in dry seasons of wet savanna and forest African ecosystems could be attributed to a longer residence time of $NH_3$ from biomass burning and agricultural waste burning sources in the atmosphere which are the main sources of atmospheric $NH_3$ in this season. Decreasing trend in ground-based $NH_3$ concentrations in the wet season at Zoétélé could be related to wet season-to-wet season increasing of the LAI ($+0.69\% \ yr^{-1}$), with a 99% significant anticorrelation of -0.57 between these two variables. Emission inventories have inherent uncertainties that may come from activity data and emission factors, or even missing emission sources. In terms of measurement data, the monthly averaged data mask considerable diurnal variability in $NH_3$ concentrations. Drivers contributing to this variability include the influence of physical and meteorological parameters, and influence of local emission sources and interactions with others atmospheric compounds on INDAAF sites.

Results reported in this paper represent the unique long-term regional characterization of ground-based $NH_3$ concentrations in Africa. Our study allows a better understanding of the main drivers of atmospheric $NH_3$ level of concentrations and trends. More field campaigns and experiments to highlight others sources (soils, livestock, fertilizer quantities, …) is still necessary before obtaining a definitive answer to decreasing trends in ground-based concentrations of $NH_3$ at the INDAAF sites. The NitroAfrica project (2023-2026) will help fill some of these gaps with valuable data on nitrogen in rural Africa. The overall objective of NitroAfrica project is to study –coupling field experiments and different modelling approaches– the relationships and retroactions between atmospheric N deposition, N cycling in the soil-vegetation system, emissions of reactive N forms by the surface to the atmosphere, atmospheric chemistry and regional climate.

## Data availability

The INDAAF NH$_3$ observations are available at https://indaaf.obs-mip.fr/ upon registration. TRMM 3B42 precipitation data are available from https://pmm.nasa.gov/data-access/downloads/trmm. The IASI NH$_3$ data are available from The IASI https://iasi.aeris-data.fr.

## Author contribution

Money Ossohou designed the study, conducted the statistical analysis, and wrote the paper.
Jonathan Hickman, and Corinne Galy-Lacaux contributed to study design and edited the paper.
Lieven Clarisse, Pierre-François Coheur, and Martin Van Damme developed the original IASI trace gas retrievals and edited the paper.
Marcellin Adon, and Véronique Yoboué edited the paper.
Eric Gardrat, and Maria Dias Alvès analysed the samples.

## Competing interests

The authors declare that they have no conflict of interest.

## Disclaimer

Publisher's note : Copernicus Publications remains neutral with regard to jurisdictional claims in published maps and institutional affiliations.

## Acknowledgement

This paper is part of the INDAAF (International Network to study Deposition and Atmospheric chemistry in Africa) long-term project supported by the CNRS/INSU (Centre National de la Recherche Scientifique / Institut National des Sciences de l'Univers), by the ACTRIS-FR research infrastructure and by the IRD (Institut de Recherche pour le Developpement). We are particularly grateful to all INDAAF local field technicians, for their work. This study has received funding from the European Union's Horizon 2020 research and innovation programme under the Marie Skłodowska-Curie Grant Agreement No. 871944. This work was supported by the CNES. IASI has been developed and built under the responsibility of the Centre National d'Études Spatiales (CNES, France). It is flown on board the Metop satellites as part of the EUMETSAT Polar System. The research in Belgium was funded by the Belgian State Federal Office for Scientific, Technical and Cultural Affairs (Prodex HIRS) and the Air Liquide Foundation (TAPIR project). This work is also partly supported by the FED-tWIN project

ARENBERG funded via the Belgian Science Policy Office (BELSPO). L. Clarisse is Research Associate supported by the Belgian F.R.S.-FNRS.

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
