# Peer review of "Trends and seasonal variability of ammonia across major biomes in Western and Central Africa inferred from long-term series of ground-based and satellite measurements"

_Atmospheric Chemistry and Physics, 2022_

## Author Comment (AC1)

**Response to reviewers**

Title: Trends and seasonal variability of ammonia across major biomes inferred from long-term series of ground-based and satellite measurements
Author(s): Money Ossohou et al.
MS No.: acp-2022-793
MS type: Research article

Dear Editor-in-Chief :

The authors would like to thank the editor and the reviewers for their valuable time and comments. They raise constructive and helpful questions and comments for improving the manuscript. We have modified the paper according to these comments, and addressed all questions raised by the two reviewers of the manuscript. More specifically, we have restructured some paragraphs and especially the results and discussion sections. We have applied new criteria based on the number of observations to the IASI $NH_3$ database. We also used anthropogenic $NH_3$ emission inventory data, discussed the results obtained taking into account physical parameters and $NH_3$ emission inventory data. We have redrawn figures and added new tables. We hope you find our manuscript suitable for publication and look forward to hearing from you.
Below, we provide a point-by-point response explaining how we have addressed each of the reviewers' comments ; note that our responses to the comments are in blue.

Yours sincerely, Money Ossohou, Jonathan E. Hickman, Lieven Clarisse, Pierre-François Coheur, Martin Van Damme, Marcellin Adon, Véronique Yoboué, Eric Gardrat, Maria Dias Alvès and Corinne Galy-Lacaux
Email : ossohoumoney@gmail.com

**Anonymous Referee #1**

Summary

The authors present an assessment of satellite-based (IASI) and ground-based (INDAAF) monitoring records of NH3 across three ecosystems in Western and Central Africa. Africa has been an understudied region for NH3 emissions due to the lack of surface monitoring infrastructure. This study attempts to provide insights on NH3 emissions and sources in this region using two independent measurements. However, there are major issues that need to be addressed. The key question this study is trying to answer does not appear clear to me. For example, it is mentioned that a prior study already found biomass burning emissions drive seasonal variation in NH3 total column densities in West Africa (Line 68-70). The authors also mention that IASI and INDAAF have been compared to each other and used to study seasonal cycles and trends of NH3 in this region (Line 75-82; Line 167-169). What new information do we gain from this study? I am also concerned about the disagreement between IASI and ground-based observations in many aspects. The scientific implications of these disagreements are never discussed. How do we reconcile contradicting measurements and use them to improve long-term NH3 assessments? These are the fundamental questions the authors need to think more carefully about.

Thank you. This manuscript represents the first effort to evaluate surface $NH_3$ variability and trends in West Africa during the entire 1998-2018 period—a period of dramatic change in the region—using any data source.
We want to emphasize that the earlier studies mentioned in lines 167-169, in which INDAAF data were used to evaluate IASI, do not represent a use of IASI + INDAAF data that would compromise the novelty of the work presented here (And note that the studies mentioned in lines 68-70 and 75-82 did

not conduct any comparison of INDAAF and IASI observations). Of the studies using IASI and INDAAF observations, Hickman et al. (2018) conducts a cursory visual comparison of monthly values over the Sahel for the single year of 2008, with no analysis of seasonality or trends. Ossohou et al. (2019) presents an analysis of NO2 and HNO3—there is no NH3 data or analysis in the paper. And Van Damme et al. (2015) is a retrieval validation effort—it's a completely different exercise than that presented here.

We further hope the value of direct surface observations is evident, and that the particular value of the measurements made by the INDAAF network are doubly so. West African ecosystems are unique, subject to dynamic seasonal variability impacted widespread biomass burning, and to rapid decadal changes in fossil fuel combustion (e.g., Liousse et al. (2014)), burning regimes (e.g. Andela & van der Werf (2014); Andela et al. (2017)), population, and agricultural extent and intensity—it is arguably the region experiencing the most dynamic change at the moment (e.g., Hickman et al. (2021)), and yet we have no studies evaluating surface concentrations of NH3 over the last decade+, or that can incorporate NH3 dynamics over the last decade or so into a longer-term trend extending to the 1990s. And unlike the United States, China, or Europe, each of which has extensive networks conducting numerous observations of surface atmospheric composition, and each of which will soon have its own geostationary monitor of atmospheric composition, West Africa is not well instrumented. INDAAF represents the only source of long-term observations of surface atmospheric composition in the region, and currently the only way to document multi-decadal change in NH3.

We also hope the reviewer agrees that there is value in analyses done with surface observations independent of satellite studies; each approach has strengths and weaknesses. IASI provides broad regional coverage, but there are also biases and uncertainties associated with IASI retrievals as there are with any satellite instrument, and seasonal variation in the number of retrievals possible in West Africa. Also like any satellite instrument, there are additional uncertainties associated with inferring surface concentrations from column measurements. Passive samplers have their own limitations (e.g. temporal resolution, spatial coverage), but they provide consistent year-round surface observations that have been the foundation of important scientific advances, whether in process or validation studies.

Specific comments

Title: I suggest modifying to "across major biomes in Western and Central Africa" or something similar to clarify that you are only looking at certain parts of Africa, and in fact, only six locations rather than the whole region.

We agreed modifying to ''across major biomes in Western and Central Africa''. The new title is : "**Trends and seasonal variability of ammonia across major biomes in Western and Central Africa inferred from long-term series of ground-based and satellite measurements**".

Line 53: Define "soil emissions" more precisely. Soil emissions of NH3, reactive nitrogen or total nitrogen?

The sentence has been clarified in **now line 58** : ''Soil emissions of NOx over …''

Line 100: Among all the INDAAF stations, is NH3 only measured at the six stations shown in Figure 1?

No, NH$_3$ is also measured at partner sites such as Tunisia (Medenine) and South Africa (Welgegund). Please, visit INDAAF website at https://indaaf.obs-mip.fr/ for more informations.

Line 136: The 14.3% reproducibility between duplicates is technically the precision of the measurements, not accuracy.

The sentence has been changed in **now line 158** to ''The precision of the measurements of passive samplers …''.

Line 140: Was there a reason for the higher data loss at Bomassa?

Logistical issues explain higher data loss in Bomassa. First, the access to the remote site of Bomassa in the heart of the equatorial forest explain sometimes delays in passive samplers' deployment by local techncians. This issue made that some of the results after analysis are not qualified. In addition, the screening of the dataset of Bomassa indicate some pollution of some samplers, or a too long delay between exposition and analysis (due to poor handling). Finally, it leads to the rejection of some samplers (28%) that is higher than for the other INDAAF sites.

Line 155: MetOp-A retired in late 2021. Did you see any worsening in instrument performance over time? If instrument performance was not an issue, why did this study only include observations up to 2018?

Conceptually, the satellite observations are provided as a supplement to the surface observations, which are the focus of the manuscript. The satellite observations, which provide retrievals of the total column, are intended to provide additional, hopefully useful, context for understanding the dynamics in surface atmospheric composition observed at the INDAAF stations, but we also consider them to be secondary to the surface observations in this paper. Given this perspective, we believe it is most appropriate to include satellite time series that match the INDAAF time series.

Line 162: You mentioned that satellite observations are for 1°x1° around each site, then why regrid to 0.25° first? Also, how many IASI observations do you usually find near the ground stations on a monthly basis? Do you see seasonal differences in the availability of satellite observations?

In the new version of the manuscript, we use the monthly time-series over 2008-2018 period considering all the IASI-A morning observation within 100 km from the INDAAF site (circle around the site, not a box anymore). The primary reason for working with 100 km data is to match other data sources (e.g. GFED and now CEDS).

Table. Mean number of IASI observations per month within a circle of 100 km of diameter centered on the location of each INDAAF site over the 2008-2018 period.

|  | Jan | Feb | Mar | Apr | May | Jun | Jul | Aug | Sep | Oct | Nov | Dec |
|---|---|---|---|---|---|---|---|---|---|---|---|---|
| **Banizoumbou** | 442 | 458 | 427 | 343 | 339 | 269 | 238 | 176 | 274 | 471 | 536 | 518 |
| **Katibougou** | 449 | 474 | 497 | 431 | 285 | 225 | 226 | 156 | 227 | 363 | 456 | 472 |
| **Djougou** | 511 | 450 | 342 | 300 | 225 | 161 | 91 | 58 | 95 | 296 | 464 | 547 |
| **Lamto** | 214 | 190 | 204 | 215 | 121 | 35 | 34 | 40 | 18 | 48 | 133 | 142 |
| **Bomassa** | 60 | 65 | 112 | 124 | 144 | 131 | 120 | 106 | 102 | 125 | 144 | 65 |
| **Zoétélé** | 96 | 68 | 75 | 70 | 62 | 23 | 20 | 13 | 19 | 33 | 82 | 98 |

The figure below shows the average annual number of IASI NH3 observations over the period 2008-2018.

[Figure]

Table 2:

- This is not really a fair comparison. 21 years of ground data could be very different from 11 years of satellite data. Your low correlation between the Katibougou station and IASI could be a result of this. The time series plots are fine, but if you want to compare the statistics, I suggest only looking at periods when IASI and ground stations overlapped (i.e., 2008-2018). You can add another table comparing ground data before 2008 and ground data after 2008.
- Correlations between ground-based and satellite observations are made using the same period, i.e. where IASI and ground stations overlapped (2008-2018). As asked by the reviewer 1, we have added another tables (**Tables 3, 5 and 7**) comparing ground-based concentrations of NH3 before 2008 and after 2008.

- It would be more meaningful if you could label the corresponding month and year where you saw the minima/maxima.
- Thank you for this observation. The authors think that label the corresponding month and year where their saw the minima/maxima in the table would make it unreadable because the same minimum concentration value (i.e. 0.7 ppb) is often observed for several months at a time. For this reason, the authors have chosen to plot figures with average annual profiles (**Fig. 3, 5 and 7**) where the readers could see these informations. The authors have also indicated the minimum and maximum in the paragraphs related to these figures.

- I noticed the minimum ground NH3 concentrations were same across all the sites (0.7 ppb), which happened to be equal to the samplers' detection limit. Do you think these just reflect the minimum detectable level, while the real minimum concentrations of NH3 in the air may be lower?
- Over the period 1998-2007, 23 analytical series of measurements have been performed using Ionic Chromatography (IC) and a total of 230 field blanks have been analyzed. Detection limits for NH3 was calculated from field blanks and found to be 0.7±0.2 ppb (Adon et al., 2010). Using the INDAAF passive sampler technique, we believe that 0.7 ppb is the minimum detectable concentration for NH3.
- The same comments apply to Tables 3 and 4.
- Thank you. We did it.

Line 217: Aren't the dashed lines for the GFED4 emissions?

Thank you for this observation. Thank you for this observation. In the new version of the manuscript, we have added an inventory of anthropogenic NH₃ emissions. To avoid showing them all on the figure (which would become overloaded), we decided to calculate correlations with these emission data. Significant correlation coefficients have been interpreted in the manuscript.

Figure 3: The three NH3 datasets (IASI, INDAAF and GFED4) have very different units, making it somewhat difficult to associate them together. I know they represent different quantities, but is there a way you can translate the units into more meaningful contexts?

We have removed GFED4 from figures 3, 5 and 7 to use the statistical approach in the manuscript.
Line 238: You only showed rainfall data among all meteorological parameters. I would like to see at least how temperature changes throughout the year in each of the climate zone. Can you show the monthly temperature since it is closely related to NH3 volatilization potential?

We acknowledge the important point raised here, and agree with reviewer 1. We have shown temperature data in **figures 3, 5 & 7**.

Line 255: I see the amount of rainfall is drastically higher later in the wet season, but I don't see any evidence telling whether it is erratic or evenly distributed within each month. Do you have data to support this?

Sahelian dry savannas (Banizoumbou, Niger and Katibougou, Mali) are characterized by a short wet season (June to September), but a long dry season (October to May). We have selected 8-year (2001-2008) of daily rainfall amount (in mm) from September 16th to October 30th at Katibougou – representing the end of the wet season, beginning of the dry season – to show that the amount of rainfall is erratic. It's important to note that rainfall amount is higher at Katibougou compared to Banizoumbou.

The dry Sahelian savanna ecosystems (Banizoumbou, Niger and Katibougou, Mali) are characterized by a short wet season (June to September), but a long dry season (October to May). We have selected 8 years (2001-2008) of daily rainfall (in mm) from September 16 to October 30 in Katibougou to show that the amount of rainfall is erratic for this ecosystem (figure below). It is important to note that the monthly rainfall distribution in Banizoumbou is more erratic than that of Katibougou. *Orange cells mean that it did not rain.*

[Figure]

Line 270: Can you provide a more quantitative overview of biomass burning events in this region, such as the average number of fires in each month in dry savannas, wet savannas and forests?

Based on MODIS observations between 2008 and 2018, about 5% of the area covered by dry savanna ecosystems burns annually, 26% of wet savanna ecosystems, 22% of northern forest/grassland mosaic ecosystems, and 2% of equatorial forest ecosystems, with most of the burning occurring during the dry season (Table Sx).

Table Sx: The fraction of annual burned area by month in select ecoregions of sub-Saharan Africa. Data represent the mean of observations from MODIS between 2008 and 2018 ; ecoregion definitions are as described in Hickman et al. 2021 (https://doi.org/10.1029/2020GB006916) and illustrated in plot Sx.

| Month | Dry_savanna | Wet_savanna | Northern_mosaic | Equatorial Forest |
|---|---|---|---|---|
| Jan | 0.00560837 | 0.06567305 | 0.07557587 | 0.00303727 |
| Feb | 0.00436254 | 0.02697975 | 0.02081838 | 0.00251232 |
| Mar | 0.00297438 | 0.00939677 | 0.00535943 | 0.00165749 |
| Apr | 0.00131618 | 0.00387437 | 0.0027148 | 0.00103541 |
| May | 0.00074776 | 0.0013802 | 0.00123126 | 0.00038196 |
| Jun | 0.00026099 | 0.00042781 | 5.33E-05 | 0.00093235 |

| | | | | |
|---|---|---|---|---|
| Jul | 5.25E-05 | 5.62E-05 | 3.47E-06 | 0.00215427 |
| Aug | 2.55E-06 | 3.79E-05 | 7.36E-06 | 0.00159459 |
| Sep | 0.00055987 | 0.0003719 | 3.90E-05 | 0.00051408 |
| Oct | 0.00989503 | 0.01269798 | 0.00094386 | 0.00013732 |
| Nov | 0.01697483 | 0.05265935 | 0.01956299 | 0.00093128 |
| Dec | 0.00950919 | 0.08854965 | 0.08919434 | 0.0030895 |

[Figure]

Plot: Ecoregion definitions used in calculating mean monthly burned area presented in Figure Sx. Definitions are derived from The Nature Conservancy's (TNC) map of Terrestrial Ecosystems (available at http://maps.tnc.org/gis_data.html) as described in Hickman et al. (2021a)

Line 284: Again, this looks like a red flag to me if IASI didn't correlate with ground-based concentrations at all, and no theory was proposed to explain why. The anticorrelation during the wet season (Figure 5a) was not explained either. There could be a highly localized source near the site biasing the ground measurements, or something could be off in your comparison methodology, which goes back to my earlier comment that you need to provide more information on how you mapped the IASI data to the ground sites.

As the reviewer notes, mismatches between station observations and satellite observations integrated across a 100 km around each site can be expected due to the very different nature of the observations. IASI provides a total column value, which we have averaged over an area of roughly 100 square miles for each station comparison. The surface stations provide a point measurement at the surface. So any differences between a) surface concentrations and concentrations at any other altitude in the atmosphere or b) between composition at the station and at any other point in the 100 square-mile area can produce a mis-match between the station observations and the IASI retrieval.

We would further note that there is large uncertainty associated with many of these measurements. For example, in the cited example (Figure 5a) the error bars for the surface observations encompass a range that includes values that would provide a seasonal pattern matching that of IASI.

Ultimately, the function of the IASI data in this manuscript is as a supplement to the station data. Although we may not be able to explain precisely why a mismatch is present, we believe it provides some useful evidence to consider when assessing the representativeness of the station for regional conditions.

Line 303: Do you mean the seasonal fluctuations are less remarkable at Djougou than Lamto?

Yes. As we can see from the annual mean profiles (figure 5), the maximum values of ground-based NH3 concentrations and IASI NH3 total columns almost always appear in February at the Lamto site, which is not the case for Djougou.

Line 337: I found the first part of this sentence confusing: "Ground-based NH3 concentrations are high or low in both the dry and wet seasons, with no clear seasonality". You can simply say that ground-based concentrations show no clear seasonality.

**Now line 383** : ''Ground-based concentrations show no clear seasonality at Bomassa and Zoétélé.''

Line 357: If biomass burning is the largest contributor to NH3 in wet savannas and forests, why did IASI and INDAAF not peak in the month where you see highest GFED4 emissions at Djougou and Lamto?

The reviewer 1 is right. Normally, IASI and INDAAF did peak in the month where we see highest GFED4 emissions in the wet savanna sites. We believe this may be due to uncertainties in the emissions data from GFED4. Indeed, uncertainty in GFED4 fire emissions stems from uncertainty in burned area, fuel consumption, and emission factors but is poorly constrained. Uncertainty in $NH_3$ emission factors is large because few fires have been sampled, which adds to the total uncertainty, burned area and fuel consumption in savannas are, in general, better constrained than in other biomes (Hickman et al., 2018). We expect that a large reason for a disagreement between the observed $NH_3$ peak and that modeled by GFED4 is because GFED4 is based on estimates of burned area from MODIS, which has recently been found to hugely underestimate the area burned by small fires (Ramo et al., 2021; Roteta et al., 2019). This underestimation occurs predominantly in the shoulder season, and precisely when we observe the INDAAF and IASI peaks. In February, total burned area in northern hemisphere sub-Saharan Arica may be underestimated by MODIS more than a factor of 2, and in March, by an order of magnitude (Ramo et al. 2021), leading to a large underestimate of emissions in GFED4. Rainfall is also higher in these months, which may contribute to greater smoldering fires and higher emissions of species that are not fully oxidized, such as NH3.

We have added anthropogenic emission data from the CEDS inventory to our analyses. The results show that residential sources and agriculture contribute significantly to $NH_3$ emissions in wet savanna and forest.

Line 359: This statement seems exactly opposite of what I see on Figure 3, where IASI and ground concentrations both reach their maxima in the early wet season (May-June) and minima in the dry season (December-January).

This sentence refers only to wet savanna and forest ecosystems. The sentence has been clarified in **now line 405**: ''Our study demonstrates that highest NH3 concentrations in wet savanna and forest ecosystems are recorded during the period when fires predominate (December-February), while the lowest are obtained when rainfall is high.''

Line 370: What are the correlations between the two NH3 datasets and GFED4 at other sites?

The following table show correlations the two NH₃ datasets (ground-based and IASI) and GFED4 at the other sites. *Values in parentheses represent p-value.*

| | Ground-based Vs GFED4 | IASI Vs GFED4 |
|---|---|---|
| **Banizoumbou, Niger** | -0.2 *(0.64)* | -0.13 *(0.13)* |
| **Katibougou, Mali** | 0.02 *(0.67)* | -0.07 *(0.37)* |
| **Djougou, Benin** | 0.05 *(0.52)* | -0.09 *(0.27)* |
| **Bomassa, Republic of congo** | -0.06 *(0.03)* | 0.18 *0.03)* |
| **Zoétélé, Cameroon** | -0.00 *(0.99)* | **0.39** *(<0.001)* |

Line 380: I suggest saying "which is not the case at Djougou" than "which is unusual at Djougou". One other thing to note is that canopy heights at Djougou and Lamto look very different on Figure 1. Dry deposition can be affected by both LAI and the vertical distance between canopy and instrument. The canopy looks much shorter at Djougou, with lots of vegetation being lower than the instrument. This could be another reason why NH3 emissions may be less intercepted by the canopy at Djougou.

We thank the reviewer for this comment. The correction has been made in **now line 437** : ''which is not the case at Djougou''.

We have also completed with the comment : ''One other thing to note is that canopy heights at Djougou and Lamto look very different on Figure 1. Dry deposition can be affected by both LAI and the vertical distance between canopy and instrument. The canopy looks much shorter at Djougou, with lots of vegetation being lower than the instrument. This could be another reason why NH3 emissions may be less intercepted by the canopy at Djougou.''

Line 402: You mentioned temperature here but you never showed any temperature analysis.

Thanks for this comment. We have added mean annual temperature profile in **figures 3, 5 and 7**. We We also found significant correlation coefficients between NH₃ and air temperature at all INDAAF sites. Many other comments have been made in the manuscript.

Line 410: Are you sure the IASI trends are in units of molec cm-2 yr-1, not x10^15 molec cm-2 yr-1?

Sorry for this mistake. We have added "x $10^{15}$" to the line indicated by reviewer 1, and made the correction throughout the document.

Line 460: This may explain the lack of correlation between IASI and ground data, but not necessarily the contrasting trends. If biomass burning was increasing over time (which supposedly would be captured better by IASI), the ground sites should show insignificant trends if they simply missed the fire plumes. However, since decreasing ground-based trends were seen at almost all sites, something else was happening. Perhaps it was the LAI that you mentioned, but we would need to see the season-to-season change in LAI at all sites to be sure.

Indeed, something was happening. Nevertheless, at the Lamto site, the study on the chemical content of rainfall showed that NH₄⁺ in rainfall shows an increasing trend (Ossohou et al., 2020). An increasing trend in rainfall, combined with the decreasing trend in burned area (Andela et al., 2017), could be expected to result in a net increase in LAI over time. A positive trend in LAI could be expected to increase N deposition locally, potentially contributing to a reduction in surface atmospheric concentrations over time.

Line 471: You used the word "more important" multiple times throughout the text when comparing the NH3 concentrations. The meaning of "important" can be misinterpreted by some. I suggest using descriptions such as "significantly higher" if you simply want to compare the values (e.g., ground based NH3 concentrations were significantly higher in dry savannas than wet savannas and forests).

Thank you. We have replaced ''more imortant'' by ''significantly higher'' throughout the text.

Line 475: I'm not sure if this conclusion holds true just based on the surface concentrations. You did not actually conduct a source attribution to quantify how much NH3 is from soils, livestock versus biomass burning.

We have modified the sentence in **now lines 574-577** : ''If we consider only ground-based measurements, the results show that $NH_3$ emissions from Sahelian soils, livestock and agriculture (only at Katibougou) in the dry savanna ecosystem lead to average concentrations in the dry season that are equal to those obtained in the wet savanna ecosystem, globally dominated by biomass burning and agriculture.''

Minor comments

Line 65: I suggest changing to "~250 Mha of land area was burned"

Done in **now line 70**.

Figure 1 caption: Remove "location" in "10 stations across Africa location"

We did it.

Line 125: No need to use quotation marks around Laboratoire d'Aérologie (LAERO)

Done in **now line 147**.

Line 163: The IASI version 3 datasets

Thanks. We have changed the sentence.

Line 190: is limited

Done in **now line 211**.

Line 238: when the weather is at its warmest and driest

Thanks. We have changed the sentence.

Line 290: Should be Table 3, not Table 2

Correction is made. Thank you.

Line 395: wet and dry seasons

Done in **now line 364**.

Line 460: so biomass burning alone could not explain

Done in **now line 549**.

Citation: https://doi.org/10.5194/acp-2022-793-RC1

**Anonymous Referee #2**

The manuscript provides valuable insights into NH3 concentrations and trends in Africa using both ground-based observations and satellite measurements. However, the manuscript simply describes NH3 seasonality, IASI &INDAAF comparison, and NH3 trends without detailed interpretations. The goal and implications need to be clarified and further analyses need to be done to illustrate the results.

Major comments:

The introduction needs to be restructured to address the following questions:

(1) Why do we care about NH3 seasonality? Is NH3 seasonality highly uncertain?

We thank the reviewer for this valuable questions. To make more clear informations about NH3 seasonality, we have added some informations in the introduction. Here are these informations :

"Due to its high reactivity, a significant fraction of the NH3 emitted is rapidly deposited within a 1 km radius of the source (Fowler et al., 1998). It is clear that the seasonal distributions of NH3 vary depending on the dominant source type and remains a very important element in understanding local emission sources and changing in environmental conditions (Tang et al., 2018)."

"INDAAF measurements are of great interest to remove some uncertainties in order to understand the seasonality of several trace gases including $NH_3$ in Western and Central Africa. Some uncertainties are caused by the scarcity of data on the spatial and temporal distribution of application of synthetic fertilizers and animal manure by crop, and the prevailing management conditions (Beusen et al., 2008)."

(2) You only briefly mentioned biomass burning emissions without explaining the importance, e.g., biomass burning accounts for how much NH3 emissions in Africa? What is the impact on air quality and radiative forcing?

Thank you ! This is a good point arised here. We have adressed the questions in **now lines 72-74** : "Biomass burning emits large amounts of aerosols and trace gases which significantly affect biosphere-atmosphere interface, atmospheric chemistry, cloud properties, Earth radiation budget, global carbon cycle, ecosystem and biodiversity, air quality and atmospheric circulation (Crutzen and Andreae, 1990; Andreae and Merlet, 2001; Stocker et al., 2013)."

(3) Line 163 – 169 should be moved to the introduction.

We have moved lines 163-169 to the introduction in **now lines 85-92**.

IASI & INDAAF comparison:

(1) Is there any difference between the agreement for WS and DS? There may be fewer IASI measurements in WS, and hence worse agreement compared to DS.

Here is the table that we have presented to the Reviewer 1 about the mean number of observations per month at each INDAAF site.

Table. Mean number of IASI observations per month within a circle of 100 km of diameter centered on the location of each INDAAF site over the 2008-2018 period.

| | Jan | Feb | Mar | Apr | May | Jun | Jul | Aug | Sep | Oct | Nov | Dec |
|---|---|---|---|---|---|---|---|---|---|---|---|---|
| **Banizoumbou** | 442 | 458 | 427 | 343 | 339 | 269 | 238 | 176 | 274 | 471 | 536 | 518 |
| **Katibougou** | 449 | 474 | 497 | 431 | 285 | 225 | 226 | 156 | 227 | 363 | 456 | 472 |
| **Djougou** | 511 | 450 | 342 | 300 | 225 | 161 | 91 | 58 | 95 | 296 | 464 | 547 |

| Lamto | 214 | 190 | 204 | 215 | 121 | 35 | 34 | 40 | 18 | 48 | 133 | 142 |
| Bomassa | 60 | 65 | 112 | 124 | 144 | 131 | 120 | 106 | 102 | 125 | 144 | 65 |
| Zoétélé | 96 | 68 | 75 | 70 | 62 | 23 | 20 | 13 | 19 | 33 | 82 | 98 |

From this table, we can see that there is fewer IASI observationsin the wet savanna compared to dry savanna. Globally, the number of observations decrease from the dry savanna to forest ecosystems (Figure below).

[Figure]

(2) Figure 3, 5, and 7: add monthly temperature and number of IASI pixels.

We have added monthly temperature to the **figures 3, 5 & 7**.

We used new satellite datasets (not very different from the first one). In the new approach, we have taken into account monthly observations greater than 12 for our analyses. The area considered is a circle 100 km in diameter centered around each INDAAF site (to remain consistent with NH3 emissions data). The average number of observations is shown in the table and graph above.

(3) Table 2, 3, and 4: add IASI & INDAAF agreement.

We are not quite sure we understood the point. Nevertheless, we have added INDAAF to the titles of tables 2, 3 and 6, which have now become tables 2, 4 and 6.

(4) What are the implications of using IASI data in Africa? Can IASI represent the ground truth in all major African ecosystems?

Thank you for this important point raised here.

Limited availability of ammonia ($NH_3$) observations is currently a barrier for effective monitoring of the nitrogen cycle. It prevents a full understanding of the atmospheric processes in which this trace gas is involved and therefore impedes determining its related budgets. Since the end of 2007, the Infrared Atmospheric Sounding Interferometer (IASI) satellite has been observing $NH_3$ from space at a high spatio-temporal resolution. This valuable data set, already used by models, still needs validation. One of the implications of using IASI data in Africa is its validation with INDAAF data, which is unique on the African continent.

This study provides an analysis of variability and trends in surface atmospheric composition, and INDAAF is literally the only data source available to understand this question-there simply are no other

datasets that can be used to evaluate the temporal variation of surface atmospheric composition in West Africa, a region that is undergoing rapid and dynamic changes in emissions.

Ultimately, the function of the IASI data in this manuscript is as a supplement to the station data. Although we may not be able to explain precisely why some mismatches are present, we believe it provides some useful evidence to consider when assessing the representativeness of the station for regional conditions in Africa. Globally, IASI data follow ground profiles when cloud cover is low, but also when plumes evolve at low altitudes. According to this study, we can assume that IASI $NH_3$ data represent well the ground truth during 1) all the seasons of the year in the dry savanna and 2) the dry season in the wet savanna.

Trend analysis:

(1) It is unfair to compare the trends for different time domains. Can you add a figure for 2008 – 2018 ground $NH_3$ trends and comparison with IASI trends? I eyeballed Figure 4 and for Lamto the decrease from 1998 to 2008 is significant and that might have driven the long-term trends for 1998 – 2018.

The objective of this study is not to compare trends of ground-based and satellite data. We did not actually compare INDAAF to IASI $NH_3$ trends, and we are not aware of any studies that compare INDAAF observations to $NH_3$ trends in any detail, or with recent data.

The variability of the $NH_3$ concentration time series led us to perform other statistical tests including the homogeneity test. The results are presented in answer to your next question.

(2) It would be interesting to see if the decreasing trends are slowing down or in transitions to increasing trends during the past decade.

To see if the eventual decreasing trends are slowing down or in transitions to increasing trends, we applied homogeneity tests for all sites with 1000 Monte Carlo simulations (Pettitt's Test). Homogeneity tests have shown that only the ground-based data are not homogeneous, i.e. there is a date from which there is a change in the data. A change in variation was observed in 11/2014, 10/2011, 08/2010, 02/2003, 02/2009 and 02/2003 in Banizoumbou, Katibougou, Djougou, Lamto, Bomassa and Zoétélé, respectively (figures below).

[Figure]

[Figure]

[Figure]

[Figure]

After Pettitt's tests, we have applied the Mann-Kendall tests to time series of more than 10 years (Table below).

| Sites | ppb month$^{-1}$ |
|---|---|
| Banizoumbou (01/98-11/2014) | 0.06 (p<0.01) |
| Katibougou (01/98-10/2011) | -0.07 (p=0.17) |
| Djougou (07/05-08/2010) | < 10 yrs |
| Lamto (02/03-12/18) | -0.01 (p=0.38) |
| Bomassa (01/98-02/2009) | +0.08 (p=0.97) |
| Zoétélé (02/2003-12/18) | +0.02 (p=0.94) |

(3) Add trend lines for Figure 2, 4, and 6.

Thanks for the comment. We did it.

(4) GFED only provides wildfire emissions. For the emission trends, use EDGAR or other inventories that provide total NH$_3$emissions.

We thank the reviewer for this suggestion. In the revised version of the manuscript, we use CEDS emission inventories in our analyses.

(5) Any thoughts on the contribution from trends in meteorology and the partitioning between NH$_3$ and NH$_{4+}$?

Yes, our study shows for example that the increasing trend of air temperature at Djougou, Lamto and Bomassa is significantly correlated with IASI $NH_3$ at these sites. Air temperature at these sites therefore contributes to IASI $NH_3$ trends.

No, our study did not enable us to understand the partitioning between $NH_3$ and $NH_4^+$. However, a previous study at Lamto showed increasing trend in $NH_4^+$ concentrations in rainwater (Ossohou et al., 2020).

Minor comments:

Line 41: (Bouwman et al., 2002a) delete brackets and the comma.

Done in **now line 43**

Line 48: PM2.5 subscript.

Done in now line 50.

Line 68: wrong citation. Van Damme et al. 2018 does not focus on wildfire emissions.

Sorry, we change the citation to "van der Werf et al., 2017" in **now line 77**.

Line 78: (Van Damme et al., 2021) delete brackets and the comma.

Done in **now line 94**.

Line 79: change 2.3%.decade-1 to 2.3%·decade-1.

Done in **now line 98**.

Line 124: (Ferm, 1991) delete brackets.

Done in **now line 146**.

Line 175: $NO_2$ data?

Sorry, it's a mistake. **Now line 195**, we have deleted $NO_2$.

Line 333: change second largest to second largest terrestrial.

Done in now line 399.

Line 388: NH3 subscript.

Done in **now line 451**.

Line 451: change 0.56 to $0.56 \times 10_{15}$.

Thank you. We have corrected this type of error throughout the manuscript.

Line 471: change more important to higher.

We have changed "more" by "higher" in the manuscript.

Line 740: duplicates for Whitburn et al. 2015a and 2015b.

The right reference is Whitburn et al., 2015. We have corrected this mistake in the manuscript.

Citation: https://doi.org/10.5194/acp-2022-793-RC2

**References**

Adon, M., Galy-Lacaux, C., Yoboué, V., Delon, C., Lacaux, J. P., Castera, P., Gardrat, E., Pienaar, J., Al Ourabi, H., Laouali, D., Diop, B., Sigha-Nkamdjou, L., Akpo, A., Tathy, J. P., Lavenu, F., and Mougin, E.: Long term measurements of sulfur dioxide, nitrogen dioxide, ammonia, nitric acid and ozone in Africa using passive samplers, Atmospheric Chemistry and Physics, 10, 7467–7487, https://doi.org/10.5194/acp-10-7467-2010, 2010.

Andela, N., Morton, D. C., Giglio, L., Chen, Y., van der Werf, G. R., Kasibhatla, P. S., DeFries, R. S., Collatz, G. J., Hantson, S., Kloster, S., Bachelet, D., Forrest, M., Lasslop, G., Li, F., Mangeon, S., Melton, J. R., Yue, C., and Randerson, J. T.: A human-driven decline in global burned area, Science, 356, 1356–1362, https://doi.org/10.1126/science.aal4108, 2017.

Andreae, M. O. and Merlet, P.: Emission of trace gases and aerosols from biomass burning, Global Biogeochemical Cycles, 15, 955–966, https://doi.org/10.1029/2000GB001382, 2001.

Beusen, A. H. W., Bouwman, A. F., Heuberger, P. S. C., Van Drecht, G., and Van Der Hoek, K. W.: Bottom-up uncertainty estimates of global ammonia emissions from global agricultural production systems, Atmospheric Environment, 42, 6067–6077, https://doi.org/10.1016/j.atmosenv.2008.03.044, 2008.

Crutzen, P. J. and Andreae, M. O.: Biomass Burning in the Tropics: Impact on Atmospheric Chemistry and Biogeochemical Cycles, Science, 250, 1669–1678, https://doi.org/10.1126/science.250.4988.1669, 1990.

Fowler, D., Sutton, M. A., Smith, R. I., Pitcairn, C. E. R., Coyle, M., Campbell, G., and Stedman, J.: Regional mass budgets of oxidized and reduced nitrogen and their relative contribution to the nitrogen inputs of sensitive ecosystems, Environmental Pollution, 102, 337–342, https://doi.org/10.1016/S0269-7491(98)80052-3, 1998.

Hickman, J. E., Dammers, E., Galy-Lacaux, C., and van der Werf, G. R.: Satellite evidence of substantial rain-induced soil emissions of ammonia across the Sahel, Atmospheric Chemistry and Physics, 18, 16713–16727, https://doi.org/10.5194/acp-18-16713-2018, 2018.

Hickman, J. E., Andela, N., Tsigaridis, K., Galy-Lacaux, C., Ossohou, M., Dammers, E., Van Damme, M., Clarisse, L., and Bauer, S. E.: Continental and Ecoregion-Specific Drivers of Atmospheric $NO_2$ and $HNO_3$ Seasonality Over Africa Revealed by Satellite Observations, Global Biogeochemical Cycles, 35, https://doi.org/10.1029/2020GB006916, 2021.

Ossohou, M., Galy-Lacaux, C., Yoboué, V., Adon, M., Delon, C., Gardrat, E., Konaté, I., Ki, A., and Zouzou, R.: Long-term atmospheric inorganic nitrogen deposition in West African savanna over 16 year period (Lamto, Côte d'Ivoire), Environ. Res. Lett., 16, 015004, https://doi.org/10.1088/1748-9326/abd065, 2020.

Ramo, R., Roteta, E., Bistinas, I., van Wees, D., Bastarrika, A., Chuvieco, E., and van der Werf, G. R.: African burned area and fire carbon emissions are strongly impacted by small fires undetected by coarse resolution satellite data, Proc. Natl. Acad. Sci. U.S.A., 118, e2011160118, https://doi.org/10.1073/pnas.2011160118, 2021.

Roteta, E., Bastarrika, A., Padilla, M., Storm, T., and Chuvieco, E.: Development of a Sentinel-2 burned area algorithm: Generation of a small fire database for sub-Saharan Africa, Remote Sensing of Environment, 222, 1–17, https://doi.org/10.1016/j.rse.2018.12.011, 2019.

Stocker, T. F., Qin, D., and et al.: Climate Change 2013 : The Physical Science Basis. Intergovernmental Panel on Climate Change, Working Group I Contribution to the IPCC Fifth Assessment Report (AR5), Cambridge University Press, Cambridge, United Kingdom and New York, NY, USA, https://doi.org/10.1017/CBO9781107415324, 2013.

Tang, Y. S., Braban, C. F., Dragosits, U., Dore, A. J., Simmons, I., van Dijk, N., Poskitt, J., Dos Santos Pereira, G., Keenan, P. O., Conolly, C., Vincent, K., Smith, R. I., Heal, M. R., and Sutton, M. A.: Drivers for spatial, temporal and long-term trends in atmospheric ammonia and ammonium in the UK, Atmos. Chem. Phys., 18, 705–733, https://doi.org/10.5194/acp-18-705-2018, 2018.

van der Werf, G. R., Randerson, J. T., Giglio, L., van Leeuwen, T. T., Chen, Y., Rogers, B. M., Mu, M., van Marle, M. J. E., Morton, D. C., Collatz, G. J., Yokelson, R. J., and Kasibhatla, P. S.: Global fire emissions estimates during 1997–2016, Earth Syst. Sci. Data, 9, 697–720, https://doi.org/10.5194/essd-9-697-2017, 2017.

---

## Author Response (AR2)

**Response to reviewers**

Title: Trends and seasonal variability of ammonia across major biomes inferred from long-term series of ground-based and satellite measurements
Author(s): Money Ossohou et al.
MS No.: acp-2022-793
MS type: Research article

Dear Editor-in-Chief :

The authors would like to thank the editor and the reviewer for this second round of revision of our manuscript. Statistical tests applied to the manuscript have shown that ground-based $NH_3$ concentrations at INDAAF sites are decreasing, while vertical $NH_3$ total column densities from IASI show increasing trends. The contrasting trends between INDAAF and IASI are due to the spatial extent of the measurements. INDAAF measurements are located at a precise point, while IASI covers a much wider area (7,854 square km). As result, IASI also takes into account $NH_3$ emissions (1) far from the INDAAF site and (2) at altitude. We hope you find this second round of review suitable for publication and look forward to hearing from you.
Below, we provide a point-by-point response explaining how we have addressed each of the reviewer comments. Note that our responses to the comments are in blue.

Yours sincerely, Money Ossohou, Jonathan E. Hickman, Lieven Clarisse, Pierre-François Coheur, Martin Van Damme, Marcellin Adon, Véronique Yoboué, Eric Gardrat, Maria Dias Alvès and Corinne Galy-Lacaux
Email : ossohoumoney@gmail.com

**Anonymous Referee**

I would like to thank the authors for taking the time to address the reviewers' comments. I am mostly satisfied with the authors' response and the revised manuscript, but I do have a few more comments:

• Line 75: Why do you say the number from Levine (1996) is the best guess, and what is the uncertainty of the estimate? Do you get the same number if you sum up the emissions in GFED4?

We sincerely appreciate this valuable comment. We believe that this reference to global $NH_3$ emissions from biomass burning needs to be updated. We have therefore replaced it with a more up-to-date reference ''*Bray et al. (2021)*''.

**In now line 74** : ''Recently, Bray et al. (2021) estimated average $NH_3$ emissions from biomass burning at a global scale over the period 2001-2015 at $4.53\pm0.51$ Tg $yr^{-1}$.''

• Line 310: The fact that monthly averaged IASI total columns can be negative is confusing. Maybe add a note to explain why you can get negative numbers from IASI.

**Now in line 191** : ''It is important to note that monthly IASI NH3 total columns can be negative. The negative columns are related to measurement noise, inherent to any type of measurement.

In the ANNI product, noise is translated both to positive and negative columns, unlike some other measurement products that translate noise always to positive columns, resulting in positive biases (*Whitburn et al., 2016*). On average, such noise averages out over long time periods, resulting in a column close to zero over remote regions. The few negative monthly values that are observed here, are close to zero, and occur during months with low ammonia and low measurement sensitivity. Note that these few negative values do not drive the observed trends.''

*Whitburn, S., Van Damme, M., Clarisse, L., Bauduin, S., Heald, C. L., Hadji-Lazaro, J., Hurtmans, D., Zondlo, M. A., Clerbaux, C., and Coheur, P.-F.: A flexible and robust neural network IASI-NH₃ retrieval algorithm: New IASI-NH₃ NN Retrieval Algorithm, Journal of Geophysical Research: Atmospheres, 121, 6581–6599, https://doi.org/10.1002/2016JD024828, 2016.*

• Line 424: It doesn't look like either the r or p-value between "Ground-based and GFED4" is significant enough to say that biomass burning has an influence on ground-based concentrations at Bomassa and Zoétélé.

Thank you for your comment. We want to say that « These results show that NH3 emissions from biomass burning influence ground-based concentrations of $NH_3$ in Lamto, and total column densities of $NH_3$ in the Lamto, Bomassa and Zoétélé areas » We have modified the sentence in **now line 431** to be more clear.

• Line 445: Since you mentioned satellite observations are influenced by a lot of non-local dynamics, I think a 100 km diameter may be too large for the purpose of comparing satellite and ground-based trends. This is probably why you keep seeing a lack of correlation between IASI and ground stations throughout the analysis. Have you tested how sensitive the correlation is with respect to the diameter you choose (100 km vs 50 km vs 25 km, for example) ?

We thank the reviewer for this valuable comment. Indeed, a diameter of 100km centered around each site could be large, but the sentivity test (*table below*) shows that :

- This diameter does not hide any significant correlation
- We obtain the lowest percentage of data that does not meet our selection criterion (number of observations >= 20)

| Site (diameter of the circle centered on the site) | Correlation coefficient (*p-value*) | Percentage of missing values (counter<20) |
|---|---|---|
| Banizoumbou (100km) | **0.28 (<0.01)** | 0 |
| Banizoumbou (50km) | **0.31 (<0.01)** | 12 |
| Banizoumbou (25km) | 0.29 (*0.04*) | 42 |
| | | |
| Katibougou (100km) | 0.06 (*0.5*) | 0 |
| Katibougou (50km) | 0.1 (*0.25*) | 5 |
| Katibougou (25km) | 0.07 (0.5) | 50 |
| | | |
| Djougou (100km) | 0.03 (*0.7*) | 2 |

| | | |
|---|---|---|
| Djougou (50km) | 0.08 (*0.4*) | 16 |
| Djougou (25km) | 0.01 (0.9) | 54 |
| | | |
| Lamto (100km) | **0.55 (*<0.01*)** | 14 |
| Lamto (50km) | **0.60 (*<0.01*)** | 41 |
| Lamto (25km) | - | 80 |
| | | |
| Bomassa (100km) | 0.18 (*0.07*) | 2 |
| Bomassa (50km) | 0.18 (*0.14*) | 39 |
| Bomassa (25km) | - | 100 |
| | | |
| Zoétélé (100km) | **0.34 (*<0.01*)** | 27 |
| Zoétélé (50km) | **0.51 (*<0.01*)** | 32 |
| Zoétélé (25km) | - | 99 |

• Line 529: Now I'm a little concerned about the decreasing trends you get from ground-based concentrations given all the evidence in this paragraph showing NH3 emissions should increase over the years. This suggests to me that maybe the ground stations are not capturing the overall trend that is occurring in the region, while IASI does capture it to a certain extent. Moving forward, what do you think can be done to improve the assessment of NH3 concentrations and trends at the surface? Having more ground stations certainly will help answer these questions better.

Indeed, to better assess ground-based $NH_3$ concentrations and trends, we need more measurements in the areas of INDAAF sites. In addition, Chemistry transport models (CTMs) can be used to estimate and predict $NH_3$ concentrations at the surface, but also to assess $NH_3$ trends over a given period. The results of the INDAAF program can be used as input data for these models.

In Lamto (Côte d'Ivoire), we plan to install active $NH_3$ analyzers to better compare ground-based data and IASI $NH_3$.

• Line 555: I suggest keep the units consistent (avoid switching between km and miles), and SI units (km) are generally preferred.

Sorry for this switching. We have replaced 100 squre miles to 7,854 square km in **now line 562**.

• Line 579: You made corrections to these correlation coefficients earlier in the text and I believe it should be Lamto (r=0.55), Bomassa (r=0.18) and Zoétélé (r=0.34).

We apologize to the reviewer. We have made the corrections in **now line 586**.